# Metabolic adaptability in metastatic breast cancer by AKR1B10-dependent balancing of glycolysis and fatty acid oxidation

Antoinette van Weverwijk[1,4], Nikolaos Koundouros[2,3], Marjan Iravani [1], Matthew Ashenden[1], Qiong Gao [1], George Poulogiannis[2,3], Ute Jungwirth [1,5] & Clare M. Isacke [1]

The different stages of the metastatic cascade present distinct metabolic challenges to tumour cells and an altered tumour metabolism associated with successful metastatic colonisation provides a therapeutic vulnerability in disseminated disease. We identify the aldo-keto reductase AKR1B10 as a metastasis enhancer that has little impact on primary tumour growth or dissemination but promotes effective tumour growth in secondary sites and, in human disease, is associated with an increased risk of distant metastatic relapse. AKR1B10[High] tumour cells have reduced glycolytic capacity and dependency on glucose as fuel source but increased utilisation of fatty acid oxidation. Conversely, in both 3D tumour spheroid assays and in vivo metastasis assays, inhibition of fatty acid oxidation blocks AKR1B10[High]-enhanced metastatic colonisation with no impact on AKR1B10[Low] cells. Finally, mechanistic analysis supports a model in which AKR1B10 serves to limit the toxic side effects of oxidative stress thereby sustaining fatty acid oxidation in metabolically challenging metastatic environments.

[1] The Breast Cancer Now Toby Robins Research Centre, The Institute of Cancer Research, London SW3 6JB, UK. [2] Department of Cancer Biology, The Institute of Cancer Research, London SW3 6JB, UK. [3] Division of Computational and Systems Medicine, Department of Surgery and Cancer, Imperial College London, London SW7 2AZ, UK. [4] Present address: Division of Tumor Biology & Immunology, The Netherlands Cancer Institute, Plesmanlaan 121, 1066 CX Amsterdam, The Netherlands. [5] Present address: Department of Pharmacy & Pharmacology, Centre for Therapeutic Innovation, University of Bath, Bath BA2 7AY, UK. Correspondence and requests for materials should be addressed to C.M.I. (email: clare.isacke@icr.ac.uk)

A defining characteristic of primary tumour cells is an ability to alter their metabolism, which provides the energy and metabolites required to sustain survival in nutrient and oxygen limiting conditions. In disseminating tumour cells this need for an altered metabolism becomes more acute, as cells have to avoid anoikis-mediated cell death in the circulation and face the challenge of surviving at the metastatic site before establishment of a productive metastatic colony. Moreover, different metastatic sites pose distinct metabolic challenges to the tumour cell[1,2]. In breast cancers, these altered metabolic dependencies are now being defined but the molecular mechanisms regulating this metabolic adaptability have yet to be identified.

Here we report the analysis of a syngeneic in vivo RNAi screen to identify putative metastasis enhancers. Among the top hits from the screen was the aldo-keto reductase, *Akr1b8*. Akr1b8, and its human orthologue AKR1B10[3] are NADPH-dependent enzymes that can reduce a variety of carbonyl substrates[4]. These include the conversion of retinal to retinol[5,6] resulting in decreased retinoic acid signalling, conversion of the isoprenyl aldehydes farnesal and geranylgeranal to farnesol and geranylgeranol[7] generating precursors for protein prenylation and the reduction of cytotoxic aldehydes[8]. Although *AKR1B10* expression is upregulated in a variety of cancers including hepatocellular[9,10], lung[11], pancreatic[12] and breast[13,14], the mechanism by which elevated levels of AKR1B10 enhances metastasis is not known. We demonstrate that AKR1B10$^{High}$ cells are characterised by a reduced glycolytic capacity and an increased utilisation of fatty acid oxidation (FAO), and that this altered metabolism is required for successful colonisation of secondary sites but not primary tumour growth or metastatic dissemination.

## Results

**Akr1b8/AKR1B10 promotes breast cancer metastasis.** To identify novel enhancers of breast cancer metastasis we analysed a syngeneic in vivo shRNA screen, focusing on shRNAs that were significantly under-represented in the 4T1-Luc tumour-bearing lungs of BALB/c mice compared to preinoculation 4T1-Luc mouse mammary carcinoma cells (Fig. 1a; see Methods section). Eighty-one shRNAs were found to be significantly depleted ($Z$-score $< -2$) in the metastatic lung samples (Fig. 1b) and were then filtered by removing shRNAs that (a) did not align to the predicted target gene, (b) were significantly depleted in less than 3 of the 4 biological replicates, (c) targeted genes with expression in the lowest 50th percentile based on gene expression profiling of 4T1 cells directly isolated from tumours[15], and (d) when comparing the preinoculation cells to the initial plasmid library (Fig. 1a) showed a significant difference in abundance ($Z$-score $> 2$ or $< -2$) indicating an effect on cell viability. Filtering resulted in a shortlist of 23 significantly depleted shRNAs targeting genes encoding putative metastasis enhancers (Fig. 1c; Supplementary Table 1). Within this shortlist are known regulators of breast cancer progression and metastasis such as matrix metallopeptidase 9 (*Mmp9*)[16], cathepsin D (*Ctsd*)[17], insulin-like growth factor 1 (*Igf1*)[18] and MET (*Met*)[19], as well as inhibitors of apoptosis such as BCL2-like 2 (*Bcl2l*), BCL2-associated athanogene 1 (*Bag1*), nucleolar protein 3 (*Nol3*) and protein kinase C eta (*Prkch*), providing confidence in the ability of the screen to uncover novel metastatic regulators.

Of particular interest was the presence of the metabolic enzyme aldo-keto reductase 1b8 (*Akr1b8*) in this shortlist. The human orthologue of *Akr1b8*, *AKR1B10*, has been reported to be upregulated in a number of cancer types including breast cancer[13,14], but the clinical and metabolic consequences of this altered expression have not been investigated. First, 4T1-Luc cells were transduced with lentiviral constructs containing empty vector (shCTRL), a non-targeting shRNA (shNTC) or two independent shRNAs targeting *Akr1b8* (shAkr1b8-4 and shAkr1b8-7) (Supplementary Fig. 1a). Consistent with the screening data, where we compared shRNA representation in the starting plasmid pools with the preinoculation cells, *Akr1b8* knockdown had no significant effect on cell viability when cultured in full medium in vitro (Supplementary Fig. 1b). By contrast, following intravenous inoculation, the two *Akr1b8* knockdown cell lines showed a significant decrease in lung colonisation as monitored by in vivo IVIS imaging, ex vivo lung weight, and quantification of tumour burden (Fig. 1d).

Although these data validate the in vivo shRNA screen, intravenous inoculation does not assess the full metastatic ability of tumour cells. Consequently, we next performed a spontaneous metastasis assay in which cells were inoculated orthotopically into 4th mammary fat pad of BALB/c mice (Fig. 1e, Supplementary Fig. 2). No differences were observed in tumour take or primary tumour weight at the end of the experiment, however, there was a notable reduction in lung metastasis in both the *Akr1b8-4* and *Akr1b8-7* knockdown groups compared to the control shNTC and shCTRL groups. In this experiment, there was no significant difference in the number of 4T1-Luc tumour cell colonies derived from arterial blood collected at necropsy (Fig. 1f) indicating that the metastatic impairment in the 4T1-Luc knockdown cells was not due to reduced survival in the circulation. Consistent with this observation, there was no significant difference in cell apoptosis between shNTC and shAkr1b8-4 cells when plated into non-adherent culture in vitro (Fig. 1g). Finally, to address whether the metastatic impairment resulted from impairment of tumour cell survival after lodging in the lung vasculature, 4T1-Luc shNTC and shAkr1b8 cells were labelled with cell tracker dyes, mixed at a 1:1 ratio and injected via the tail vein into BALB/c mice (Fig. 1h). Imaging of the lungs 1 h post-injection confirmed that equal number of cells had been inoculated. Examination of lungs 16 h post-injection revealed no significant difference between the number of control and *Akr1b8*-knockdown tumour cells retained in the lungs. Together these data indicate that *Akr1b8* expression does not impact on survival in the circulation or lodging in the vasculature but is required for efficient colonisation of tumour cells within the metastatic site.

**Expression of *AKR1B10* correlates with metastatic relapse.** To address the clinical relevance of the data obtained with the 4T1 mouse models, expression of *AKR1B10*, the human orthologue of murine *Akr1b8*[3], was analysed in human primary breast cancers present in the TCGA database. Within the intrinsic subtypes, *AKR1B10* expression is significantly higher in the HER2-enriched and basal-like breast cancers compared to luminal A and luminal B cancers (Fig. 2a) and analysis by receptor expression revealed significantly higher *AKR1B10* expression in ER− compared to ER+ breast cancers, and in HER2+ compared to HER2− breast cancers (Fig. 2a). The latter finding is consistent with a previous report that overexpression of *AKR1B10* correlates with HER2 positivity in ductal carcinoma in situ (DCIS)[20]. An equivalent pattern of *AKR1B10* expression is seen in the Neve et al. dataset derived from profiling a large panel of breast cancer cell lines[21] (Supplementary Fig. 3a). Consistent with these published data, western blot (Fig. 2b, upper panel) and RTqPCR (Supplementary Fig. 3b) analysis of a smaller panel of breast cancer cell lines revealed low levels of AKR1B10 protein and mRNA in the ER+ ZR75.1 and MCF7 lines and high levels in the basal-like BT20, MDA-MB-468 and HCC1395 lines (Fig. 2b, upper panel). For further studies, AKR1B10 was ectopically expressed in the

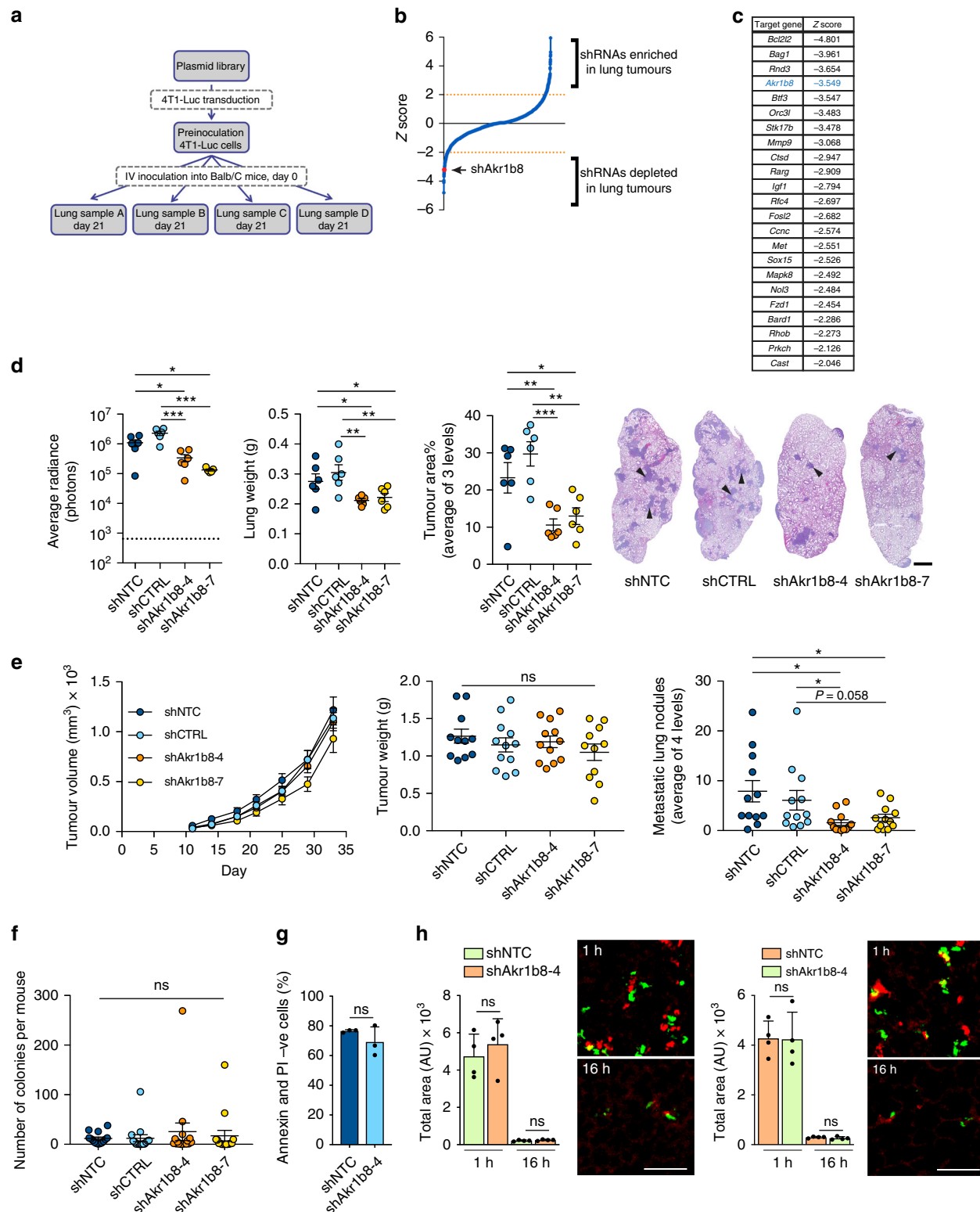

AKR1B10<sup>Low</sup> MDA-MB-231 and MDA-MB-453 lines and expression was knocked down by shRNA in the AKR1B10<sup>High</sup> HCC1395 line (Fig. 2b, lower panel). Levels of ectopically expressed protein were equivalent to that found in AKR1B10<sup>High</sup> lines, while shRNA knockdown reduced protein levels to that observed in AKR1B10<sup>Low</sup> lines.

As with the 4T1-Luc cells (Supplementary Fig. 1b), the human breast cancer cell lines with altered AKR1B10 levels showed no

difference in in vitro viability as monitored in a colony formation assay (Fig. 2c), yet when inoculated intravenously into BALB/c Nude mice, AKR1B10<sup>High</sup> MDA-MB-231 cells gave rise to a significantly increased tumour burden in the lungs compared to AKR1B10<sup>Low</sup> MDA-MB-231 cells (see vehicle-treated cohorts in Fig. 6b). Again, there was no significant difference in cell survival when cells were plated in non-adherent culture (Fig. 2d) nor in the ability of the cells to survive after lodging in the lung

**Fig. 1** Akr1b8 as a metastasis enhancer in vivo. **a** shRNA abundance was assessed in the shRNA plasmid library, shRNA transduced preinoculation cells and 4 tumour-bearing lungs samples (Samples A–D). **b** Median Z-scores for shRNA representation in the tumour-bearing lung samples compared to 4T1-Luc preinoculation cells, identifying 109 significantly enriched (Z-score > 2) and 81 significantly depleted (Z-score < -2) shRNAs. **c** Shortlist of 23 putative metastasis enhancers identified in the screen. **d** $1 \times 10^5$ 4T1-Luc transduced with two independent shRNA lentiviruses targeting Akr1b8 (shAkr1b8-4, shAkr1b8-7), a non-targeting shRNA (shNTC) or a control shRNA (shCTRL) were injected intravenously into BALB/c mice (n = 6 per group). Lung tumour burden was assessed at the end of the experiment (day 12) by ex vivo IVIS imaging (average radiance ± SEM; dashed line indicates mean radiance of aged-matched non-tumour bearing mice), ex vivo lung weight ± SEM, and quantifying % tumour area per lung section ± SEM. Representative lung sections, arrowheads indicating tumour nodules. Scale bar, 1 mm. **e** $1 \times 10^4$ 4T1-Luc cells were injected orthotopically into BALB/c mice (n = 11–12 mice per group). Animals were sacrificed on day 33. Final, tumour volume, ns for all groups except shNTC vs. shAkr1b8-7 (P = 0.005), two-way ANOVA with Tukey's multiple comparison test. Tumour weight ± SEM at necropsy. Number of lung metastases per mouse ± SEM. Representative lung images are shown in Supplementary Fig. 2. **f** From the experiment shown in **e**, quantification of mean tumour cell colonies derived from circulating tumour cells in 300 μL arterial blood per mouse ± SEM, ns for all groups. **g** 4T1-Luc anoikis assay, n = 3, mean ± SD. **h** Left panels, shNTC and shAkr1b8-4 4T1-Luc cells were labelled with CellTracker red or green dyes, respectively, inoculated intravenously into BALB/c mice. 1 and 16 h after inoculation, lungs were extracted and imaged. Data shown are mean tumour cell coverage per field of view, n = 4 mice per group per time point ± SEM. Scale bar, 100 μm. Right panels, equivalent results were obtained in a dye swap experiment. ns, not significant; *P < 0.05; **P < 0.01; ***P < 0.001; one-way ANOVA followed by two-stage step-up method of Benjamini, Krieger and Yekutieli (**d–f**), unpaired t-test (**g–h**)

vasculature (Fig. 2e), supporting the hypothesis that AKR1B10 functions to maintain efficient growth of tumour cells within the metastatic tissue.

Consistent with these findings, in a dataset of 1746 unselected breast cancers[22], high expression of AKR1B10 significantly correlated with reduced distant metastasis-free survival when considering all patients or only ER− patients. A similar trend was seen in HER2+ patients, however, the number of samples was too low to reach statistical significance (Fig. 2f). No association with outcome was seen in ER+ only patients. As AKR1B10 has been associated with chemoresistance via its ability to metabolise anticancer drugs[23], we also examined the subset of untreated patients (Supplementary Fig. 3c). Again, high expression of AKR1B10 (upper quartile) was significantly associated with reduced distant metastasis-free survival in ER−, but not ER+, breast cancer patients.

**AKR1B10$^{High}$ cancer cells have increased dependency on FAO.**
Via their oxidoreductase activity, members of the AKR family including AKR1B10 have been implicated as regulators of cellular metabolism. Aerobic glycolysis, also known as the Warburg effect, is a common feature of many cancers and characterised by increased metabolism of glucose to lactate, which is transported out of the cell resulting in local acidification. The Seahorse XF Glycolysis Stress test was used to assess glycolytic function of cells by measuring the extracellular acidification rate (ECAR) in the media (Fig. 3a). Following addition of glucose, the glycolytic rate was significantly reduced in AKR1B10$^{High}$, compared to AKR1B10$^{Low}$, breast cancer cells, as was their glycolytic capacity and glycolytic reserve. Moreover, glucose uptake was significantly reduced in all three AKR1B10$^{High}$ cell lines (Fig. 3b), indicating that AKR1B10$^{High}$ cells have a reduced requirement for glucose. Consistent with this hypothesis, in 2D culture AKR1B10$^{High}$ and AKR1B10$^{Low}$ cells showed only a modest difference in cell growth when cultured in full DMEM (4.5 g L$^{-1}$ D-glucose) but in low glucose (LG) DMEM (1 g L$^{-1}$ D-glucose) AKR1B10$^{Low}$ cells showed a significantly impaired growth rate (Fig. 3c). These data were recapitulated first in a 3D in vitro assay where AKR1B10$^{High}$ tumour spheroids showed increased growth in LG DMEM compared to the AKR1B10$^{Low}$ spheroids (Fig. 3d) and in colony forming assays where AKR1B10$^{High}$ cells were significantly more tolerant to low glucose conditions (Fig. 3e).

In addition to aerobic glycolysis, tumour cells can utilise glutamine and/or fatty acids to generate sufficient ATP and metabolites to support cellular activities. As AKR1B10$^{High}$ cells have a reduced glycolytic function, take up less glucose and are better able to survive in low glucose conditions, we used the

Seahorse XF Mito Fuel Flex Test to monitor the dependency on glutamine or fatty acids as an alternative source of energy. In none of the three cell lines was there evidence of an increased dependency on glutamine oxidation in the AKR1B10$^{High}$ cells (Fig. 4a, left panel), whereas two out of three AKRB10$^{High}$ breast cancer cell lines showed an increased dependency on fatty acid oxidation (FAO) compared to their matched AKR1B10$^{Low}$ counterparts (Fig. 4a, right panel). Moreover, AKR1B10$^{High}$ cells showed a significantly increase change in OCR (ΔOCR) following addition of the FAO substrate palmitate-BSA (Fig. 4b).

To address clinical relevance of these findings, we used a FAO 88-gene signature (FAO88; see Methods section) and demonstrated that AKR1B10 expression positively correlated with a high FAO88 score in triple negative (TN) and ER− breast cancer, but not in ER+ breast cancers (Fig. 4c) both in the TCGA dataset and in the dataset of Hatzis et al. containing 508 breast cancer patients treated with neoadjuvant chemotherapy[24]. In the Hatzis dataset there were insufficient numbers of HER2+ breast cancers for analysis, however, in the intrinsic subtype of HER2-enriched tumours high AKR1B10 expression again positively correlated with a high FAO88 score (Fig. 4c).

The processes of FAO and fatty acid synthesis are usually mutually exclusive due to their regulation by negative feedback[25]. It was notable that AKR1B10 expression in human breast cancers positively correlated with the key FAO transcriptional regulator, peroxisome proliferator-activated receptor gamma coactivator 1 alpha (PPARGC1A, also known as PGC-1α) (P = 0.009) and negatively correlated with the activators for fatty acid synthesis, acetyl-CoA carboxylase β (ACACB; P = 0.047) and acyl-CoA synthetase long chain family member 1 (ACSL1; P < 0.001), Spearman's correlation. Consistent with these clinical datasets, fatty acid synthesis as monitored by incorporation of $^{14}$C-acetate in lipids was significantly reduced in AKR1B10$^{High}$ cells (Fig. 5a). Given this negative correlation between AKR1B10 expression and lipid synthesis, AKR1B10$^{High}$ tumour cells must rely either on increased uptake of exogenous fatty acids or increased release from intracellular fatty acid stores. All three AKR1B10$^{High}$ cell lines showed increased fatty acid uptake (Fig. 5b) whereas staining of intracellular neutral lipids with the lipophilic fluorescent dye BODIPY 493/503 revealed no significant difference in lipid droplet content between AKR1B10$^{High}$ and AKR1B10$^{Low}$ cells (Fig. 5c, d), indicating that AKR1B10$^{High}$ cells predominantly fuel FAO via the uptake of free fatty acids.

**AKR1B10 sustains FAO-dependent metastatic colonisation.**
The finding that increased AKR1B10 expression promotes metastatic colonisation of the lungs (Figs. 1, 2d–f) and is

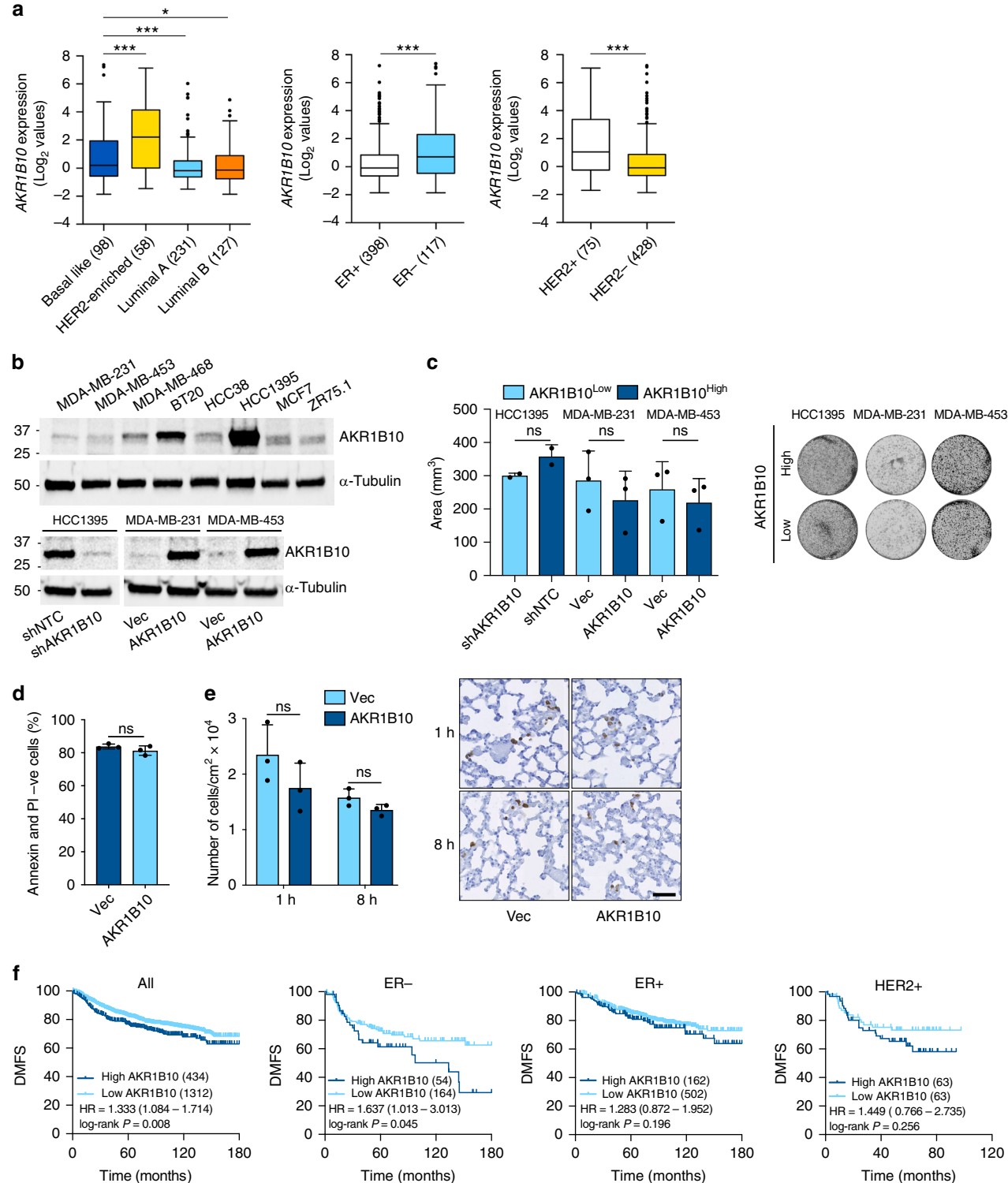

associated with an increased dependency on FAO (Fig. 4) and enhanced tolerance of low glucose culture conditions (Fig. 3c–e) raises two important questions. First, what is the mechanism by which AKR1B10 modulates these activities? AKR1B10 is distinguished from the other well-characterised AKR1B subfamily member AKR1B1 by its increased catalytic activity for retinals, isoprenyl aldehydes and, importantly, for cytotoxic aldehydes such 4-hydroxy-2-nonenal (4-HNE)[23]. The latter is a toxic lipid peroxide by-product of the elevated reactive oxygen species

(ROS) levels associated with oxidative stress. The interaction between FAO and ROS is complex. It is well documented that FAO, via its ability to generate NADPH, reduces ROS levels[26] but, conversely, it has been demonstrated that elevation of ROS in cells, for example as a result of loss of matrix attachment or treatment with rotenone or tumour necrosis factor-alpha, inhibits FAO[27,28]. Consistent with the observation that AKR1B10^Low cells have reduced viability when cultured in LG DMEM (Fig. 3c–e), glucose deprivation resulted in elevated levels of lipid

**Fig. 2** AKR1B10 is associated with increased risk of distant metastatic relapse. **a** Tukey boxplots of *AKR1B10* expression (Log$_2$ median-centred values) in the TCGA breast cancer dataset based on intrinsic subtype (one-way ANOVA with Tukey's multiple comparison test) or receptor status (*t*-test with Welch's correction). Numbers of samples in each category are indicated; *P < 0.05; ***P < 0.001. **b** AKR1B10 western blot in human breast cancer cell lines (upper panel). Lower panel, AKR1B10 expression in shNTC (AKR1B10$^{High}$) or shAKR1B10 (AKR1B10$^{Low}$) HCC1395 cells and in MDA-MB-231 cells and MDA-MB-453 cells transduced with vector-alone (AKR1B10$^{Low}$) or ectopically expressing AKR1B10 (AKR1B10$^{High}$). Molecular size markers are in kDa. Source blots are provided as Source Data File 1. **c** Colony formation assay comparing AKR1B10$^{High}$ and AKR1B10$^{Low}$ cells, n = 3 per sample ± SD. **d** 5 × 10$^4$ MDA-MB-231 cells expressing vector alone (Vec) or with ectopic expression of *AKR1B10* were cultured in 6-well low-adherence plates in 2% FBS for 24 h before annexin V and PI staining. Data shown as percent of non-apoptotic (annexin V−, PI−) cells remaining. n = 3, mean ± SD. **e** 1 × 10$^6$ MDA-MB-231 cells were injected intravenously in BALB/c Nude mice. Mice were sacrificed at 1 or 8 h (n = 3 per group per time point) and lung sections stained for human lamin A/C. Data shown is cell number per cm$^2$ ± SEM. Representative images, scale bar, 0.5 mm. ns, not significant; unpaired *t*-test (**c**–**e**). **f** Kaplan–Meier analysis of distant metastasis-free survival (DMFS) of all (n = 1746), ER+ (n = 664), ER− (218) or HER2+ (n = 126) patients in the Gyorffy et al.[22], dataset. Hazard ratios (HR) and log-rank Mantel-Cox *P*-values are shown

peroxidation as detected by BODIPY 581/591 C$_{11}$ fluorescence (Fig. 5e, f, Supplementary Fig. 4). However, lipid peroxidation levels did not increase when AKR1B10$^{High}$ cells were cultured in LG DMEM, suggesting that in metabolically challenging conditions AKR1B10 functions to limit the toxicities associated with oxidative stress and maintain FAO activity.

Second, do these cellular mechanisms operate in physiologically relevant settings? To address this, 3D tumour spheroids were treated with the FAO inhibitor etomoxir. Etomoxir had no effect on growth of AKR1B10$^{Low}$ tumour spheroids but inhibited the increased growth observed in the AKR1B10$^{High}$ tumour spheroids (Fig. 6a). More importantly, mice were inoculated intravenously with MDA-MB-231-Luc AKR1B10$^{High}$ or AKR1B10$^{Low}$ cells and, after 7 days when the tumour cells will have extravasated into the lung tissue, treated with or without etomoxir. MDA-MB-231$^{High}$ cells gave rise to a significantly increased lung tumour burden as monitored by in vivo IVIS imaging, ex vivo measurement of lung weight and quantification of metastatic burden (Fig. 6b) and this increased AKR1B10$^{High}$ metastatic colonisation was effectively impaired by etomoxir treatment, with no effect of etomoxir on the metastatic growth of AKR1B10$^{Low}$ cells.

Together these data support a model in which AKR1B10 functions to maintain FAO in tumour cells, particularly during metastatic colonisation of the pro-oxidative lung microenvironment[2].

## Discussion
The data presented here demonstrates that *AKR1B10* expression is elevated in ER− and HER2+ breast cancers and that within these breast cancer subtypes, high *AKR1B10* expression is associated with an increased incidence of metastatic relapse at secondary sites. In contrast to previous reports[29,30], we find that AKR1B10$^{High}$ breast cancer cells do not display altered survival or proliferation properties when cultured in vitro in full medium (Figs. 2c, 3c, 3e) or when grown as primary tumours in the fat pads of recipient mice (Fig. 1e). However, AKR1B10$^{High}$ cells are more successful than AKR1B10$^{Low}$ cells when cultured in nutrient poor conditions such as in low glucose (Fig. 3c, d) or when colonising the lungs (Figs. 1d, 6b) and this is associated with an increased utilisation of FAO.

The ability to modulate metabolic characteristics is a feature of metastasising tumour cells as they adapt to the unique environments that they encounter[1,2]. FAO, until recently relatively understudied in cancer, is a key source of ATP, NADH, NADPH and FADH$_2$ providing a survival advantage to tumour cells, particularly under challenging conditions such as hypoxia, nutrient stress or under therapeutic challenge[31,32]. Moreover, by analysing multiple breast cancer clinical data sets, Camarda and colleagues have demonstrated dysregulation of fatty acid

metabolism genes in triple negative (TN) compared to receptor positive breast cancers and, in particular, increased expression of key activators of FAO, such as PGC-1α (*PPARGC1A*) and decreased expression of genes encoding regulators of fatty acid synthesis[33], a pattern demonstrated here to be recapitulated in AKR1B10$^{High}$ tumours. Although the role of PGC-1α expression in metastasis is controversial[34], increased expression has been demonstrated to promote breast cancer metastasis in a variety of models systems[35–37] and to be associated with increased FAO and an enhanced ability of cells to survive in 3D acini assays[31]. Conversely, impairment of FAO decreases cell survival in acini assays[27] and reduces tumour burden in the lungs and livers following intravenous inoculation[38]. Here we demonstrate that AKR1B10$^{High}$ cells fuel FAO by an increased uptake of exogenous fatty acids. To date, the best characterised fatty acid transporters are CD36, fatty acid translocase and low density lipoprotein receptor, and it is of particular interest is the recent identification of CD36$^{bright}$ cells marking a population of metastasis-initiating cells[39], and that these cells display an upregulated FAO signature.

AKR1B10 belongs to the aldo-keto reductase (AKR) super-family of NADP(H)-dependent enzymes[4], and together with AKR1B1 and AKR1B15 form the AKR1B subfamily of enzymes characterised by their ability to reduce a variety of endogenous and xenobiotic aldehydes, dicarbonyl components and some drug ketones[23]. The *AKR1B10* gene promoter contains both an activator protein-1 (AP-1) element and an antioxidant response element (ARE)[40] and AKR1B10 expression can be regulated by AP1 downstream of IRAK1 or EGFR signalling[41,42] and by NRF2 (nuclear factor erythroid 2-related factor 2) binding to the ARE element[40,43]. Consistent with the latter, induction of oxidative stress results in NRF2-mediated upregulation of *AKR1B10* expression[44]. The lungs, due to the high levels of oxygen and exposure to toxic compounds, are characterised by a high level of oxidative stress creating a challenging microenvironment for metastasising tumour cells[2]. In addition, all metastasising tumour cells will experience oxidative stress particularly in the early stages of metastatic colonisation when they have yet to form stable cell: cell and cell:matrix attachments and experience nutrient deprivation[1,45] and, in experimental models, successful metastasis is associated with metabolic changes that permit cells to with-stand oxidative stress[46–48]. Pro-oxidative conditions result in elevated ROS production driving peroxidation of lipids that can then be degraded to reactive electrophilic lipid peroxidation products, which in turn can form covalent adducts in DNA, proteins and membrane lipids. Unchecked, these lipid peroxide breakdown products are highly damaging and cytotoxic to cells. Further, under such conditions where production of ATP via FAO would be desirable, these elevated ROS levels inhibit FAO[27,28,31]. Interestingly, there is now increasing evidence that antioxidants, which inhibit ROS production, or loss of function mutations in *Keap1*, which result in hyperactivation of the NRF2-

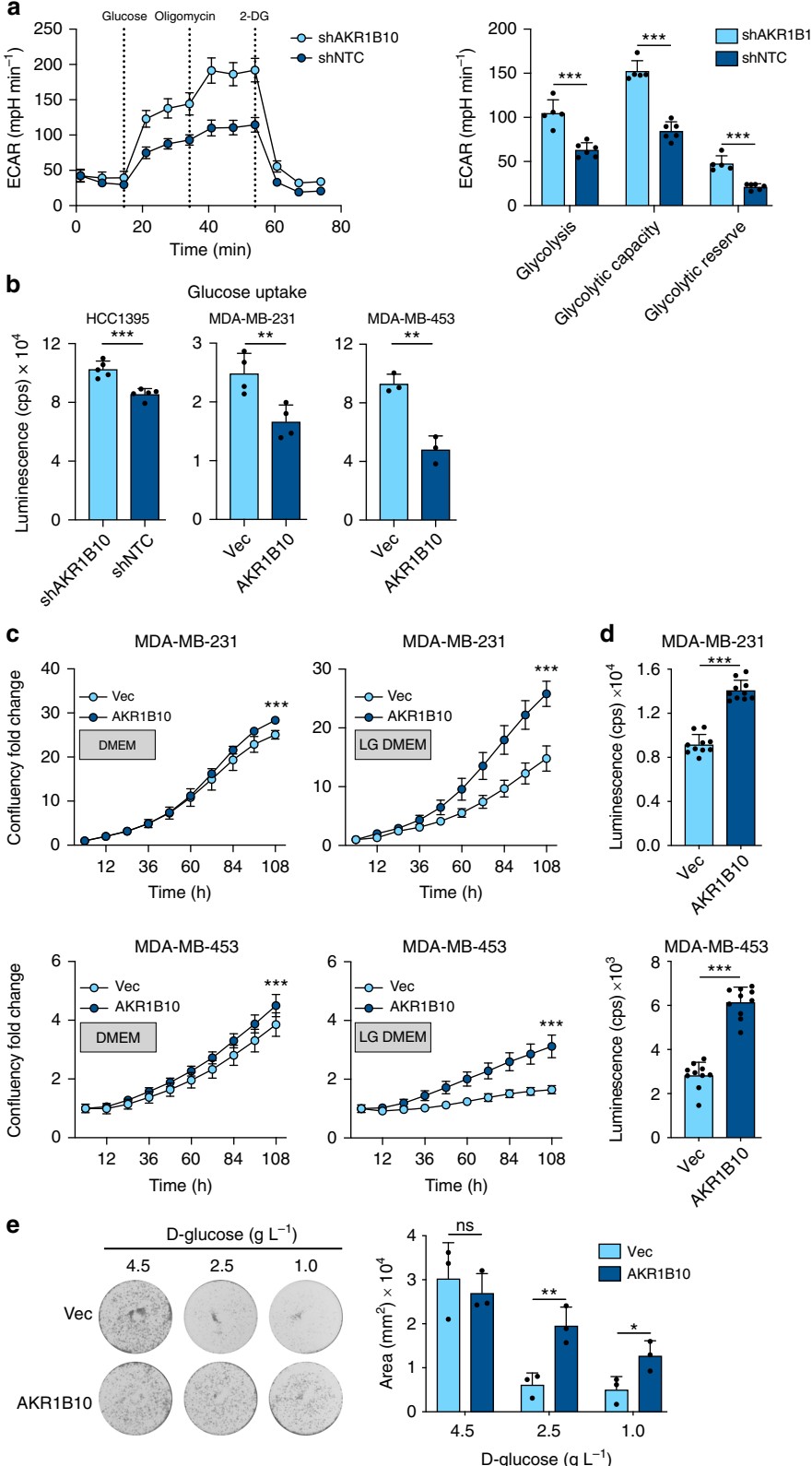

mediated endogenous antioxidant transcriptional programme, promote lung tumour progression and increase metastatic colonisation of melanomas[46–49]. These studies highlight the need for tumour cells in pro-oxidative environments to employ strategies to combat oxidative stress, and that this may be particularly pertinent for disseminated tumour cells that have yet to re-establish cell:cell and cell:matrix attachments[27,50]. Via its ability

to detoxify lipid peroxidation products by reduction of the carbonyl-groups to the corresponding alcohol metabolite[8,51–53], increased AKR1B10 activity could serve to protect tumour cells from oxidative stress-induced damage and cytotoxicity and permit maintained FAO activity. Certainly, elevated levels of FAO are associated with the increased metastasis of AKR1B10[High] cells as treatment of mice with the FAO inhibitor etomoxir impairs the

**Fig. 3** AKR1B10$^{High}$ cells have decreased glycolysis and increased tolerance of low glucose conditions. In all experiments AKR1B10$^{Low}$ and AKR1B10$^{High}$ cells are shown in pale and dark blue, respectively. **a** Glycolytic function assessed using the Seahorse XF Glycolysis Stress test. Glycolysis was determined following glucose (10 mM) injection and glycolytic capacity was determined after oligomycin (2 μM) injection. Glycolytic reserve was measured as the difference between the glycolytic capacity and glycolysis. Left panel, representative extracellular acidification rate (ECAR) profile for HCC1395 AKR1B10$^{Low}$ and AKR1B10$^{High}$ cells. Right panel, quantification of glycolytic function (shNTC, $n = 6$; shAKR1B10 $n = 5$) ± SD. **b** Glucose Uptake-Glo assay (see Methods section). $n = 4$ per sample ± SD. **c** $1 \times 10^3$ MDA-MB-231 or $5 \times 10^3$ MDA-MB-453 cells seeded into 96-well plate in DMEM or low glucose (LG) (1 g L$^{-1}$) DMEM and live imaged over 4.5 days (IncuCyte S3). $n = 10$ per condition normalised to day 0 ± SD; ***, $P < 0.001$ two-way ANOVA with Bonferroni post-testing. **d** $5 \times 10^3$ cells seeded into U-bottomed plates in LG DMEM ($n = 10$). Tumour spheroid viability analysed using CellTiter-Glo on day 6 ± SD. Equivalent results were obtained in two independent experiments. **e** $2 \times 10^3$ MDA-MB-231 cells were seeded into a 6-well plate ($n = 3$ wells per condition) and cultured in the presence of 4.5, 2.5 or 1 g L$^{-1}$ D-glucose. Data represents cell area from three independent experiments ± SD. ns, not significant; *$P < 0.05$; **$P < 0.01$; ***$P < 0.001$. Unpaired $t$-test (**a**, **b**, **d**, **e**)

enhanced metastatic growth of AKR1B10$^{High}$ cells whilst having no impact on AKR1B10$^{Low}$ metastasis.

In conclusion, the experimental and clinical data presented support a role for AKR1B10 in promoting metastasis of breast cancers functioning to support an altered metabolic programme during secondary site colonisation. The findings raise the opportunity to use AKR1B10 expression to identify breast cancer patients with an increased risk of distant metastatic relapse, and further develop AKR1B10[23] and FAO[32] inhibitors in the advanced breast cancer setting. This is particularly pertinent given that ER− breast cancer patients in general have a poorer prognosis and limited therapeutic options, making appropriate stratification for targeted therapies to control disease burden of paramount importance.

## Methods
All animal work was carried out under UK Home Office Project licences 70/7413 and P6AB1448A (Establishment License, X702B0E74 70/2902) and was approved by the Animal Welfare and Ethical Review Body at The Institute of Cancer Research. All animals were monitored on a daily basis by staff from the ICR Biological Service Unit for signs of ill health.

**Cells**. 4T1 cells were obtained from ATCC in 2013 and transduced with firefly luciferase lentiviral expression particles (Amsbio, LVP326) to generate 4T1-Luc cells. HEK293T cells and human breast cancer cell lines (MDA-MB-231, MDA-MB-453, MDA-MB-468, BT20, HCC38, HCC1395) were obtained from ATCC, and MDA-MB-231-Luc from (SibTech Inc.) between 2005 and 2012 and short tandem repeat tested every 4 months (STEmElite ID System; Promega). Subtype assessment using the absolute assignment of breast cancer intrinsic molecular subtype (AIMS) assigned MDA-MB-231 and HCC1395 cells as 100% probability of basal-like and MDA-MB-453 cells as a 100% probability of HER2-enriched. All cell lines were used within ten passages after resuscitation and were routinely subject to mycoplasma testing. HCC1395 cells were cultured in Roswell Park Memorial Institute (RPMI) medium. All other cell lines were cultured in Dulbecco's Modified Eagle's Medium (DMEM). Culture media were supplemented with 10% foetal bovine serum (FBS; Invitrogen), 50 U mL$^{-1}$ penicillin and 50 U mL$^{-1}$ streptomycin. Where indicated cells were cultured in full DMEM (4.5 g L$^{-1}$ D-glucose) or low glucose (LG) DMEM (1 g L$^{-1}$ D-glucose).

4T1-Luc cells were transduced with Mission shRNA lentiviral particles (Supplementary Table 2) at multiplicity of infection of 5. Stably transduced cells were selected in 2.5 μg mL$^{-1}$ puromycin. HCC1395 cells were transduced with GIPZ shRNA lentiviruses (Dharmacon; Supplementary Table 3), selected in 2.5 μg mL$^{-1}$ puromycin and FACSorted for GFP positive cells. For ectopic expression of *AKR1B10*, MDA-MB-231 and MDA-MB-453 cells, cells were transduced with empty vector or AKR1B10 pReceiver-Lv156 lentivirus (Genecopoeia; Supplementary Table 3) and selected in 2.5 μg mL$^{-1}$ puromycin. The cells were cultured for an additional 3 passages in selective medium to enrich the infected cell population.

**In vivo shRNA screen**. As previously detailed[54] the screen was performed with 4T1-Luc cells transduced in 24 subpools (each subpool containing 96 shRNAs) with the miR-30-based shRNA library targeting the Cancer 1000 mouse gene set[55] and inoculated intravenously into female BALB/c mice. On day 21, lungs were removed at necropsy and gDNA extracted from preinoculation cell pellets and tumour bearing lungs. shRNA representation in the original library plasmid DNA, preinoculation 4T1-Luc cells and 4 independent metastatic lung samples (samples A–D) per subpool was assessed by next generation sequencing (Fig. 1a).

shRNA representation in the preinoculation 4T1-Luc cells was compared to the representation in the 4 lung samples. Hits were defined as shRNAs that had

decreased representation (Z-score > −2) in ≥3 lung samples compared to the preinoculation cells and had no significant effect on viability when comparing shRNA representation in the plasmid pool to the preinoculation cells.

**In vivo studies**. Six to eight-week-old female BALB/c or BALB/c Nude (CAnN-Cg-Foxn1$^{nu}$/Crl) mice were purchased from Charles River. For experimental lung metastasis assays, $1 \times 10^5$ 4T1 or $1 \times 10^6$ MDA-MB-231 cells in 100 μL PBS were injected via the lateral tail vein. Where indicated, mice were randomised into two groups on day 7 and treated with etomoxir (Tocris) at 40 mg kg$^{-1}$ or vehicle (water) intraperitoneally every other day. At termination, lungs were IVIS imaged ex vivo, weighed, formalin-fixed and paraffin-embedded. Three to four micrometre thick sections were cut and stained with haematoxylin and eosin (H&E). Total number of individual nodules was counted manually in 3–4 lung sections, approximately 150 μm apart, per animal. Where indicated, lung metastatic area was quantified as the mean percentage of the area of the metastatic nodules normalised to the total lung area. For spontaneous metastasis assays, $1 \times 10^4$ 4T1-Luc cells in 50 μL PBS were injected into the 4th mammary fat pad of female BALB/c mice. Tumour growth was measured twice a week using callipers up to a maximum diameter of 17 mm. Tumour volume was calculated using the following formula: Volume = $0.5236 \times$ diameter[3]. At the end of the experiment, orthotopic tumours and lungs were harvested at necropsy. Where indicated, 300 μL arterial blood was isolated by cardiac puncture and 50 μL per well plated in DMEM plus 10% FCS in a 6-well plate per mouse. Tumour cell colonies were stained 14 days later with crystal violet. Plates were scanned at 300 dpi on EpsonV700 scanner and total number of colonies counted per mouse.

For lung retention assays, 4T1-Luc shNTC and shAkr1b8-4 cells were labelled with CellTracker Red CMTPX or Green CMFDA dyes (Molecular Probes), trypsinised, mixed at a 1:1 ratio and a total of $0.7 \times 10^6$ cells injected intravenously into BALB/c mice. Mice were sacrificed at 1 and 16 h post injection and 6 images per lung taken on a Zeiss LSM 710 microscope (×20 lens). Tumour cell colonisation within the lung was quantified in Fiji, by converting red and green images into separate binary images and measuring total tumour cell coverage per field of view. Alternatively, $1 \times 10^6$ MDA-MB-231-Luc cells were injected intravenously into BALB/c Nude mice. Mice were sacrificed at 1 and 8 h post injection and lung sections stained for human lamin A/C. Number of lamin A/C positive cells were quantified using Fiji in whole lung sections.

**Metabolic assays**. For all assays, $1.5 \times 10^4$ (MDA-MB-453) or $2.0 \times 10^4$ (HCC1395, MDA-MB-231) cells were seeded in XF96 cell culture plates incubated in a 5% $CO_2$ incubator at 37 °C overnight and results were normalised to cell number using CyQuant DNA staining (ThermoFisher).

Seahorse XF Glycolysis Stress Test. Culture medium was replaced with 175 μL pH $7.4 \pm 0.1$ bicarbonate-free DMEM supplemented with 2 mM L-glutamine, and the plate incubated at 37 °C for 1 h in a non-$CO_2$ incubator. ECAR was measured using the Seahorse XF Glycolysis Stress Test Kit (Agilent) on an XFe96 Analyzer. Final concentrations of 10 mM glucose, 2 μM oligomycin and 100 mM 2-deoxyglucose (2-DG) were used for all conditions. Glycolysis, glycolytic capacity and glycolytic reserve were calculated as follows:

Glycolysis = (maximum rate measurement before oligomycin injection) − (final rate measurement before 2-DG injection)

Glycolytic capacity = (maximum rate measurement after oligomycin injection) − (final rate measurement before glucose injection)

Glycolytic reserve = (glycolytic capacity) − (glycolysis)

Seahorse XF Mito Fuel Flex Test. Culture medium was replaced with 180 μL pH $7.4 \pm 0.1$ bicarbonate-free DMEM supplemented with 10 mM glucose, 1 mM sodium pyruvate and 2 mM L-glutamine, and the plate incubated at 37 °C for 1 h in a non-$CO_2$ incubator. OCR was measured using the Seahorse XF Mito Fuel Flex Test Kit (Agilent) on an XFe96 Analyzer. In the Mito Fuel Flex Test the import of three major metabolic substrates, fatty acids, glutamine and/or pyruvate is inhibited using etomoxir (4 μM), BPTES (3 μM) and UK5099 (2 μM), respectively. To measure dependency on FAO, OCR is measured at baseline, following injection of etomoxir (Treatment 1) and following injection of BPTES and UK5099 (Treatment 2). To measure dependency on glutamine oxidation, OCR is measured

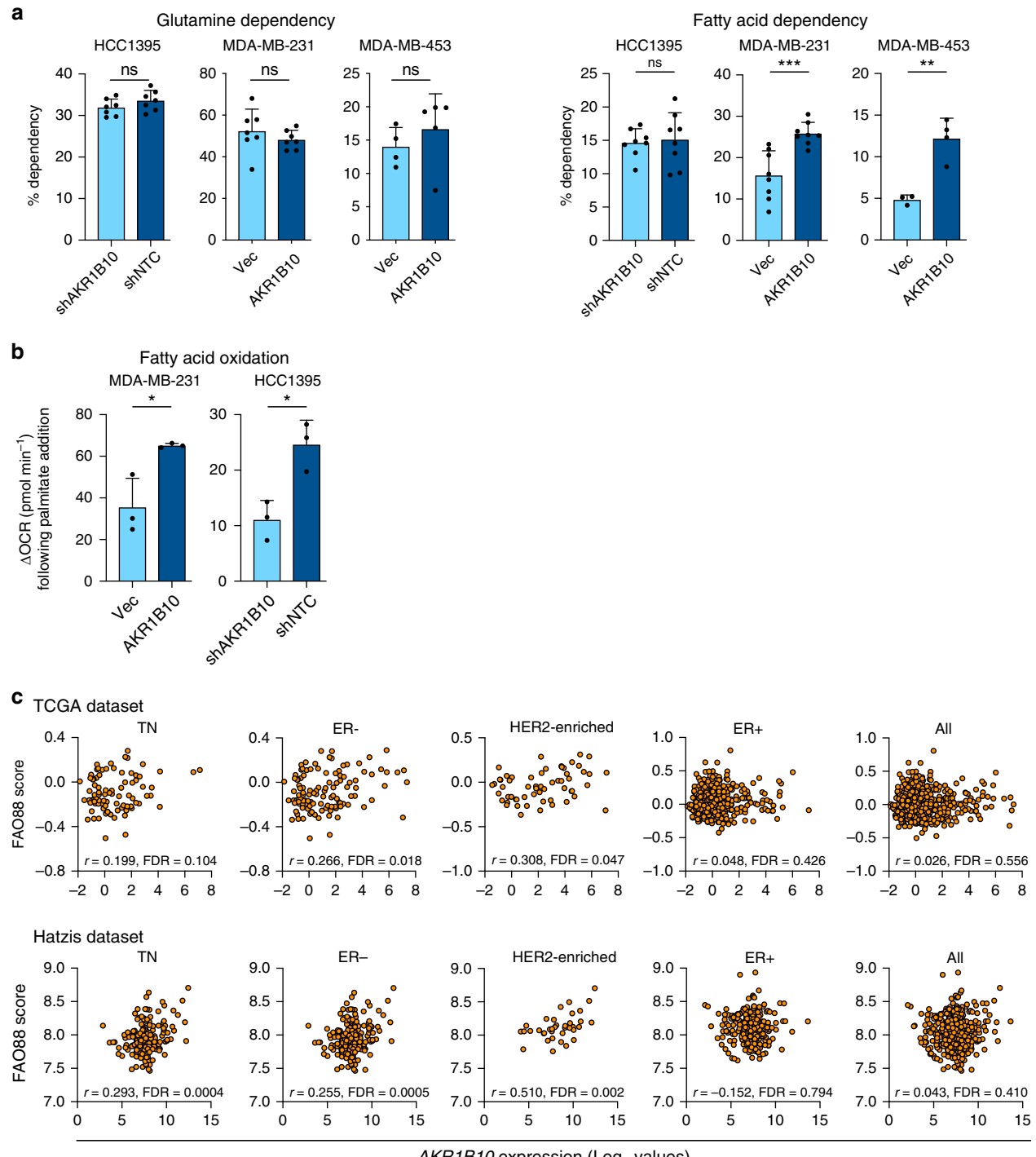

**Fig. 4** Fatty acid oxidation in AKR1B10[High] cells and breast tumours. **a** Dependency of cells on glutamine (left panel) and fatty acid (right panel) oxidation monitored using the Mito Fuel Flex test (see Methods). HCC1395 and MDA-MB-231, $n = 8 \pm$ SD; MDA-MB-453, $n = 3$–$4 \pm$ SD. **b** Change in OCR ($\Delta$OCR) following palmitate-BSA addition calculated as (OCR at the time of palmitate-BSA injection—final basal OCR). $n = 3 \pm$ SD. **c**, Pearson correlation of *AKR1B10* expression and FAO88 score in triple negative (TN), ER−, HER2-enriched, ER+ and all breast cancers in the TCGA and Hatzis et al.[24] datasets. To determine false discovery rates (FDR), the Pearson correlation *P* values were adjusted using the Benjamini-Hochberg method for multiple comparisons in each independent breast cancer clinical cohort. ns, not significant; *$P < 0.05$; **$P < 0.01$; ***$P < 0.001$, unpaired *t*-test (**a**, **b**)

at baseline, following injection of BPTES and following injection of etomoxir and UK5099. Dependency was calculated with the following formula. Dependency (%) = [(baseline OCR − Target 1 inhibitor OCR)/(Baseline OCR − All 3 inhibitor OCR)] × 100.

Palmitate-BSA FAO assay. Culture media was replaced with 175 µL of pH 7.4 ± 0.1 Krebs-Henseleit Buffer (111 mM NaCl, 4.7 mM KCl, 1.25 mM CaCl$_2$, 2 mM MgSO$_4$, 1.2 mM NaH$_2$PO$_4$) supplemented with 2.5 mM glucose, 0.5 mM carnitine and 5 mM HEPES, and the plate was incubated at 37°C for 1 h in a non-CO$_2$ incubator. 30 µL of

1 mM palmitate-BSA substrate (Agilent) was loaded directly into port A of a Seahorse loading sensor cartridge. OCR was measured at baseline and following palmitate-BSA addition on an XFe96 Analyzer. Levels of FAO are calculated as follows: Change in OCR ($\Delta$OCR) = (OCR following palmitate-BSA addition − OCR at baseline).

**Cell-based assays**. *Colony formation assay*: 0.2–5 × 10⁴ cells were seeded per well in a 6-well plate. Seven to ten days post seeding, plates were stained with crystal

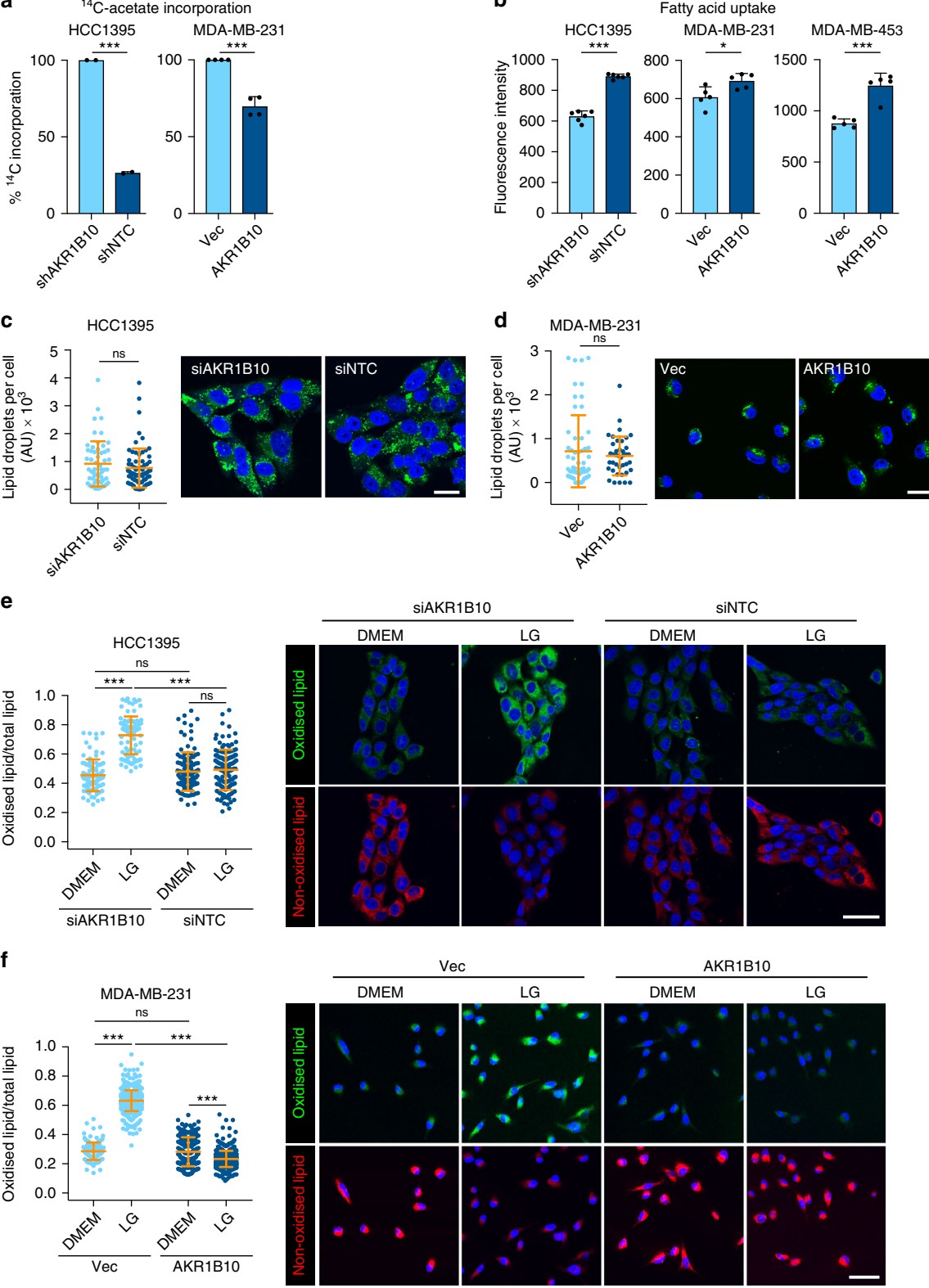

**Fig. 5** AKR1B10 limits lipid peroxidation. **a** $^{14}$C-acetate incorporated into lipids. Data from three independent experiments relative to AKR1B10$^{Low}$ cells. **b** Fatty acid uptake. $n = 5 \pm$ SD. **c** HCC1395 and **d** MDA-MB-231 cells labelled with BODIPY 493/503. Data shows lipid droplet content per cell analysed in 3–5 fields of view ± SD. Scale bar, 25 μm. **e** HCC1395 and **f** MDA-MB-231 cells cultured in DMEM or LG DMEM and stained with BODIPY 581/591 C11. Quantification of oxidised (green) BODIPY probe as a ratio of total probe (green plus red) per cell in 4–5 fields of view per sample ± SD. Representative images; scale bar, 50 μm. ns, not significant; *$P < 0.05$; ***$P < 0.001$, unpaired $t$-test (**a–d**). ns, not significant; ***$P < 0.001$, two-way ANOVA with Bonferroni post-testing (**e**, **f**)

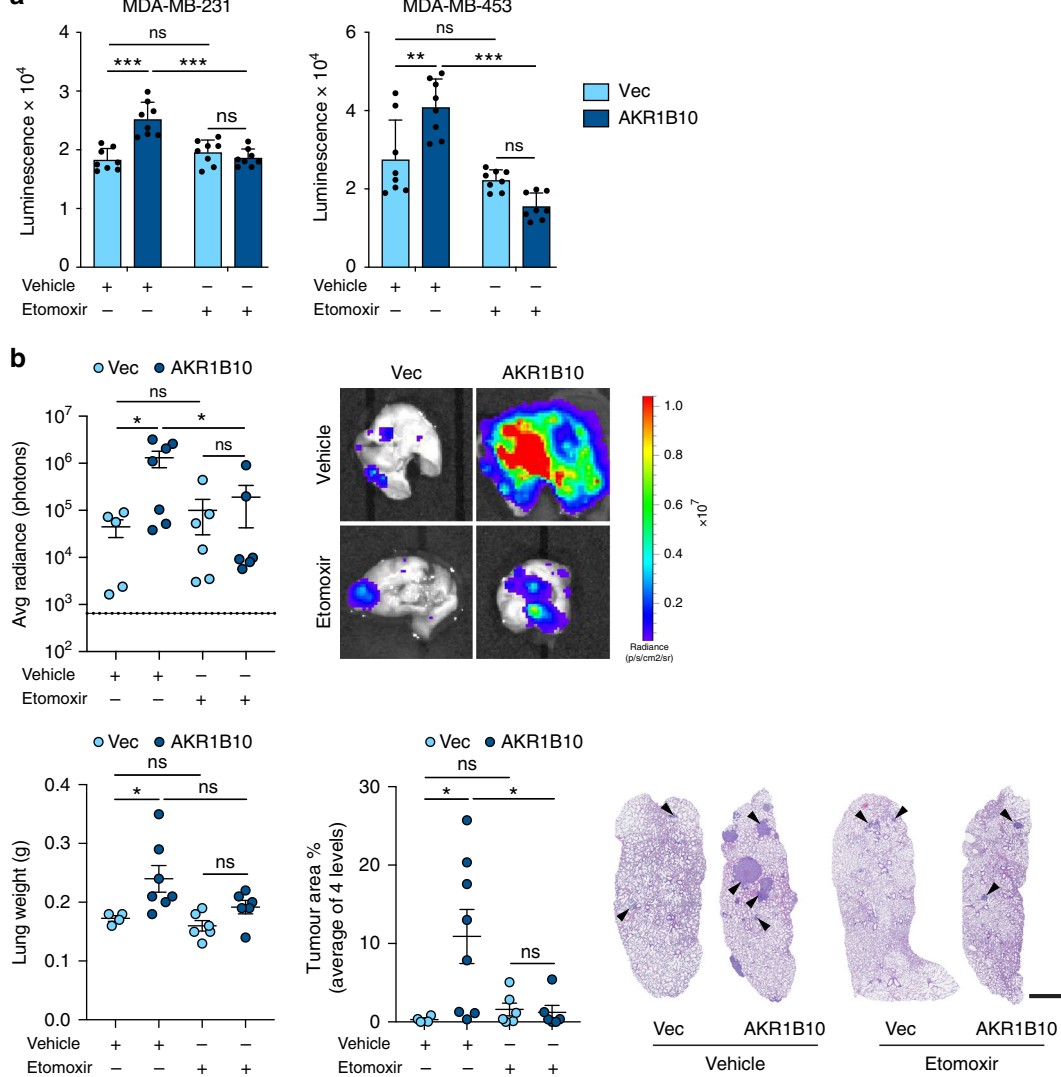

**Fig. 6** Etomoxir treatment blocks AKR1B10-mediated metastasis. **a** $5 \times 10^3$ cells seeded into ultra-low adherence U-bottomed plates. Twenty-four hours post seeding 200 μM etomoxir or vehicle was added and tumour spheroid viability assessed on day 6. $n = 8 \pm$ SD. Equivalent results were obtained in two independent experiments. **b** BALB/c Nude mice injected intravenously with $1 \times 10^6$ cells and treated with vehicle or etomoxir ($n = 5$–7 per group). Lung tumour burden was assessed at the end of the experiment (day 55) by ex vivo IVIS imaging (dashed line indicates average radiance of aged-matched non-tumour bearing mice), lung weight and the mean % tumour area from three sections cut through the lungs 150 μm apart $\pm$ SEM. Representative ex vivo IVIS images and lung sections are shown. Scale bar, 1 mm. ns, not significant; *$P < 0.05$; **$P < 0.01$; ***$P < 0.001$; two-way ANOVA with Bonferroni post-testing (**a**, **b**)

violet, dried and scanned at 1200 dpi using the GelCount colony counter system. Images were analysed using Fiji.

*Cell viability assay*: $1 \times 10^2$ cells in 100 μL medium were seeded per well in a 96-well plate and incubated at 37 °C. Immediately after seeding, and every 24 h afterwards, cells viability was analysed by CellTiter-Glo (Promega). Fold change was calculated relative to the plate read at seeding (time 0). For 3D viability assays, $5 \times 10^3$ cells were seeded into ultra-low adherence U-bottomed 96-well plates (Corning). On day 6 tumour spheroids were lysed in CellTiter-Glo for 30 min and viability analysed using a Victor X5 plate reader. Where indicated, etomoxir (200 μM) or vehicle (DMSO) was added 24 h after seeding.

*2D cell growth*: $1 \times 10^3$ cells were seeded per well in a 96-well plate and subject to live cell imaging (IncuCyte S3—Essen Bioscience) every 12 h for 4.5 days. Phase contrast images were collected, a confluence mask applied to the segmented images and analysed using IncuCyte integrated confluence algorithm).

*Anoikis assay*: $5 \times 10^4$ cells per well were seeded into low-adherence 6-well plates (Costar) in DMEM containing 2% FBS. At 24 h post-seeding, cells were stained with Annexin V-APC/PI Apoptosis Detection Kit (eBioscience) and analysed using a BD Biosciences LSRII flow cytometer with FACSDIVA and FlowJo software. Cell viability was measured as a proportion of healthy (Annexin-negative, PI-negative) cells.

*Glucose uptake assay*: $1$–$5 \times 10^4$ cells were seeded in 100 μL culture medium containing 10% FBS into a 96-well plate and incubated for 24 h at 37 °C. Cells were washed twice with PBS before the Glucose Uptake-Glo assay (Promega) was performed according to the manufacturer's protocol.

*Fatty acid uptake assay*: $1$–$3 \times 10^4$ cells were seeded in 100 μL culture medium containing 10% FBS into a 96-well plate and incubated for 24 h at 37 °C. Cells were washed and serum deprived for 1 h before 100 μL Free Fatty Acid solution (Abcam) was added. After incubating the cells for 1 h at 37 °C the fluorescence signal was measured using a fluorescence microplate reader at Ex/Em = 485/515 nm.

*Lipid synthesis assay*: Cells were incubated for 4 h in medium containing 10 μCi mL$^{-1}$ [1-$^{14}$C] acetic acid, lysed in 0.5% Triton X-100 and lipids extracted by successive addition of 2 mL methanol, 2 mL chloroform, and 1 mL $H_2O$. Phases were separated by centrifugation at 1000×$g$ for 15 min. The organic (lower) phase was recovered and dried overnight. Lipids were dissolved in Ultima Gold LSC Cocktail and counted on a scintillation counter.

*Lipid droplet analysis*: $7.5 \times 10^3$ cells were seeded on coverslips in DMEM plus 10% FBS. Twenty-four hours post seeding cells were serum-starved for 1 h, stained for 10 min with BODIPY 493/503 dye (D3922, Molecular Probes) and DAPI (Molecular Probes), and imaged on a Leica SP2 confocal microscope. Images were analysed using basic algorithms in the Columbus analysis software

package (PerkinElmer) with lipid droplets quantified using the spot finder application.

For lipid peroxidation analysis cells cultured for 48 h in DMEM or LG DMEM plus 10% FBS, stained for 30 min with $1\,\mu g\,mL^{-1}$ BODIPY 581/591 C11 (D3861, Molecular Probes) and DAPI and imaged on a Leica SP2 confocal microscope. Images were analysed using basic algorithms in the CellProfiler software package (cellprofiler.org) to quantify oxidised (green) and non-oxidised (red) BODIPY probe.

Antibodies, and the dilutions used, are detailed in Supplementary Table 4.

**Analysis of human datasets.** Series matrix files for TCGA 522 primary breast cancer samples and a neoadjuvant chemotherapy-treated invasive breast cancer clinical cohort (Hatzis, accession code GSE25066)[24] were downloaded from [https://tcga-data.nci.nih.gov/docs/publications/brca_2012/] and the Gene Expression Omnibus (GEO) site, respectively. Intrinsic molecular subtypes and clinical receptor status of ER, PGR and HER2 were retrieved from the supplemental tables of the corresponding publications. In the Tukey boxplots, box indicates the ends of the 1st and 3rd quartiles, bar indicates median, whiskers indicated 1.5 IQR (interquartile range), and dots indicate outliers. Clinical relevance of variable AKR1B10 expression was assessed using publicly available data from Gyorffy et al.[22] Unless otherwise stated, for Kaplan–Meier analysis the highest quartile of gene expression was used to dichotomise the breast cancers. For association of AKRB10 expression and FAO pathway activity, a mouse FAO gene set was obtained from [http://www.informatics.jax.org/go/term/GO:0019395]. Human orthologues of the 88 (FAO88) genes were identified using [http://www.informatics.jax.org/homology.shtml]. In Hatzis dataset, Affymetrix Human Genome U133A Array annotation file (GEO accession code GPL96) was used to map the symbol to the corresponding Affymetrix Probe_Set_ID. When multiple Probe_Set_IDs mapped to the same symbol, the Probe_Set_ID with the highest variance across samples was selected to represent the gene. Genes were discarded from further analysis if they were not mapped to either the annotation file or the expression data. FAO88 score was calculated as a mean of the normalised Log2-expression of the matched individual genes within the FAO88. Pearson correlation was used to assess associations between AKR1B10 expression and this FAO88 score of the samples in each subset. AKR1B10 expression in human cell lines was assessed in the dataset of Neve and colleagues (ArrayExpress with accession number E-TABM-157).

**Statistical analysis.** Statistics were performed using GraphPad Prism 7. Unless stated otherwise, all numerical data is expressed as the mean ± standard deviation (SD) for in vitro assays and ± standard error of mean (SEM) for in vivo tests. Comparisons between two groups were made using the two-tailed, unpaired Student's $t$-test. For experiments with two control groups (e.g., shNTC and shCTRL groups) and two experimental groups (e.g., shAkr1b8-4 and shAkr1b8-7 groups) comparisons between an individual control and an individual experimental group were made using one-way analysis of variance (ANOVA) followed by the two-stage step-up method of Benjamini, Krieger and Yekutieli. Comparisons between multiple groups with independent variables were made using two-way ANOVA with Bonferroni post-testing, with a confidence interval of 95% for individual comparisons. To determine false discovery rates (FDR) in breast cancer datasets, the Pearson correlation $P$ values were adjusted using the Benjamini-Hochberg method for multiple comparisons in each independent breast cancer clinical cohort. Statistical significance was defined as: $*P < 0.05$; $**P < 0.01$; $***P < 0.001$; ns, not significant.

**Reporting summary.** Further information on research design is available in the Nature Research Reporting Summary linked to this article.

## Data availability
Details of the datasets analysed in this study are included in the section of Analysis of human datasets. The source data (uncropped immunoblots) underlying Fig. 2b are provided as Source Data File 1.

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

## Acknowledgements

This work was funded by Breast Cancer Now (CTR-Q4-Y3), working in partnership with Walk the Walk to CMI. NK is supported by an Institute of Cancer Research studentship. The GP laboratory is supported by the Institute of Cancer Research and a Cancer Research UK Grand Challenge award (C59824/A25044). We acknowledge NHS funding to the NIHR Biomedical Research Centre at The Royal Marsden and the ICR. The results published here are in whole or part based upon data generated by the TCGA Research Network [http://cancergenome.nih.gov/]. We thank Syed Haider and his team in the Breast Cancer Now Toby Robins Research Centre Bioinformatics Core Facility, the Breast Cancer Now Toby Robins Research Centre Nina Barough Pathology Core Facility and ICR FACS and Light Microscopy Facility for support in this project. We thank Andrea Morandi (University of Florence) and Frances Turrell for reading draughts of the manuscript, David Vicente and Rebecca Orha for technical assistance and David Robertson, Adam Tyson and Alexis Barr for help with the image analysis.

## Author contributions

Experimental data was generated by Av.W., N.K., M.I., M.A. and U.J. Bioinformatics analysis was performed by Q.G. Av.W., M.I., G.P., U.J. and C.M.I. devised and oversaw the project. Av.W., M.I., U.J. and C.M.I. wrote the manuscript with input from all other authors. G.P. and C.M.I. acquired the funding.

## Additional information

**Competing interests:** The authors declare no competing interests.

