## [Peer Review File · Nature Communications]

Reviewers' Comments:

Reviewer #1:

Remarks to the Author:

The manuscript by van Weverwijk et al. describes a novel role for AKR1B10 (Akr1B8 in mice) during lung metastasis of breast cancer. The authors first identified this gene in an elegant in vivo shRNA screen for genes that facilitate lung metastasis of 4T1 mouse mammary carcinoma cells: depletion of Akr1B8 was found to reduce lung metastatic burden and similar results are reported for several known metastasis-promoting genes, thereby validating the approach. The investigators report consistent results upon depletion of the human orthologue AKR1B10 in breast cancer lines that express high levels, & reciprocal data upon overexpression of AKR1B10 in human breast cancer cells that express low levels of this gene. BC cells expressing high levels of AKR1B10 exhibit reduced glycolysis and elevated fatty acid oxidation, survive better in low glucose and have reduced evidence of cytotoxic lipid oxidation in low glucose. Etoximir treatment of mice injected with mammary tumour cells eliminates the apparent lung colonization advantage conferred by overexpression of AKR1B10.

While the potential identification of AKR1B10 as a metastasis promoter does appear to be novel, this is not the first study to suggest that oxidative stress limits metastasis (eg. Piskounova et al. Nature 2015) and the mechanism by which AKR1B10 achieves the metabolic switch from glycolysis to FAO is not addressed. Moreover, I have substantial reservations about the quality of analysis and interpretation of the results.

Major concerns:

- 1) I am not convinced that measuring total lung weight is an accurate reflection of lung metastatic burden. Besides variation from one mouse to the next, influx of blood during sacrifice can significantly alter tissue mass, especially in the lung. Although supposedly representative images are provided, lung tumour nodule distribution is never even throughout the tissue. A better way to quantify tumour burden is to section through the entire tissue, measuring tumour area every 100 or so microns. This concern is somewhat offset by the luminescence measurements, but the authors need to better calibrate tumour burden with luminescence.
- 2) In Figure 1D, the effect of Akr1b8 depletion is only significant against the empty vector control but not significant against the more appropriate control, which is non-targeting shRNA. This needs to be addressed
- 3) As the authors point out, tail vein-injection is at best an incomplete measure of metastatic potential, which makes the data shown in Figure 1E of paramount importance, given that this is the only measure of spontaneous metastasis in the manuscript. A significant reduction in the number of [lung metastasis] nodules is reported but as discussed in point 1 above, this appears to be an incomplete measurement. Additionally, these data need to be confirmed with a second hairpin and compared with an shNT control instead of empty vector.
- 4) In Figure 2, the analysis of AKR1B10 protein expression in human cell lines is incongruent with the analysis of mRNA expression in breast cancer subtypes. This raises a major concern as to the relevance of human expression data as presented. Does mRNA not reflect protein expression? For publication in a Nature family journal I would really expect TMA analysis of protein expression here.
- 5) The influence of AKR1B10 levels on lung colonization is only shown for 1 cell line (with the same concerns about metastasis burden as raised in [1]) and may only reflect a subclonal difference. Moreover, analysis of spontaneous metastasis needs to be included.
- 6) There is no attempt to uncover the mechanism underlying the metabolic switch achieved upon AKR1B10 overexpression

Minor Concerns:

- 1) What is the scale on the ordinate axes in Figure 2A?
- 2) For Figure 4C, False discovery rates should be presented in lieu of p values, as the analysis requires correction for multiple comparisons.

3) The scale on the ordinate axes of Figures 1D and 6B should start at 0 rather than 10^3

Reviewer #2:

Remarks to the Author:

The manuscript by Van Weverwijk et al identifies a metabolic enzyme, AKR1B10 as a novel enhancer of breast cancer metastasis from an in vivo shRNA screen. The authors demonstrate the following:

1. Depletion of AKR1B10 leads to significant decrease of metastasis to the lung in an intravenous and spontaneous metastasis assay without any effect on the primary tumor growth.
2. Cell lines with high AKR1B10 levels are better at forming tumors in the lungs compared to cells with low AKR1B10 levels in an intravenous metastasis assay, while the in vitro colony formation capacity is not affected by AKR1B10 levels.
3. Higher levels of AKR1B10 reduce the glycolytic capacity, glucose uptake and dependency of cells on glucose for survival and proliferation.
4. Increase in AKR1B10 may lead to an increase in fatty acid oxidation, although this point is not made as clearly as the rest
5. High levels of AKR1B10 are correlated with an increase of intracellular fatty acid stores and a decrease in lipid peroxidation
6. High levels of AKR1B10 sensitize metastasizing cancer cells to inhibition of fatty acid oxidation by etomoxir in vivo

Specific Comments:

The main finding of the manuscript is very interesting and important to the field of metastasis and metabolic plasticity of metastasizing cancer cells. The fact that cancer cells are able to adapt to their sites of growth during metastasis is an idea that has been gaining traction in the field. Even more importantly, metabolic adaptations during metastasis have been widely discussed in reviews, however not many specific targets that are important for metastatic spread but not primary tumor growth have been identified, which makes this work novel and important. In particular, the ability of etomoxir to block metastasis of AKR1B10-expressing cells is a very impressive and clinically relevant result!

However, there are several concerns that need to be addressed:

1. Throughout the manuscript, authors use lung weight as well as number of nodules that is determined by difference H&E staining to look at the difference in metastasis. However, the cell lines used in the manuscript are labelled with luciferase. It would be more accurate to quantify metastatic burden using bioluminescence, since in several figures including Fig 1e the difference in the number of nodules is very small, and barely statistically significant and seems to be mostly driven by two mice out of 19 that were analyzed. A more accurate quantification of changes in metastatic burden would make the phenotype a lot more convincing. It would be interesting to see if there is metastasis to other organs other than the lung but not necessary for the main point of the manuscript.
2. The authors want to address the effect of Akr1b8/AKR1B10 knockdown on survival of cancer cells in circulation (for examples in Fig 1f), however the assay used in the manuscript is not convincing and not correct. Mixing the cells with the dye and injecting them in the tail vein to see how many cells end up lodged in the lung and how many remain after 8 or 16 hours is not modeling survival in circulation, but rather most likely looking at the clearing of the cells from the lung as well as their survival in the lung, since cells go straight from the injection to the lung within seconds with minimal time in circulation. The authors have a spontaneous metastasis model that they use in the manuscript already, it would be much more convincing and accurate to look for the presence of cancer cells in the blood of those mice by flow cytometry.

3. Authors use confluency fold change in Figure 3 as a way to monitor effects of glucose depletion on cell growth. This is an unusual way to represent the results. It would be better if the data was confirmed with an alternative approach such as changes in cell number or cell survival (e.g. with Cell Titer Glo), which the authors use in other figures in the manuscript already.
4. Authors clearly show that they have both overexpression and knockdown cell lines for AKR1B10. The manuscript would much stronger and more convincing if the authors used both sets of cell lines for each of the assays shown, rather than pick and choose which one to show.

Minor comments:

1. Why are there are no error bars on the control bars in Figure 2e?
2. Figure 4 is probably the main point of the paper since it shows effects of AKR1B10 on Fatty Acid Oxidation but it is very difficult to follow even for someone familiar with the assays. Better labeling and schematics would really help.
3. Generally better/more precise labelling of the axes and figures (e.g. Figure 4 and 5) would make the manuscript a lot easier to read.

Overall, this is a very interesting paper with a lot of convincing and impressive results. It could be a great contribution to the field once the major concerns are addressed.

Metabolic adaptability in metastatic breast cancer by AKR1B10-dependent balancing of glycolysis and fatty acid oxidation

Antoinette van Weverwijk, Nikolaos Koundouros, Marjan Irvani, Matthew Ashenden,
Qiong Gao, George Pouligiannis, Ute Jungwirth and Clare M. Isacke

Response to reviewers' comments

Reviewer #1

The manuscript by van Weverwijk et al. describes a novel role for AKR1B10 (Akr1B8 in mice) during lung metastasis of breast cancer. The authors first identified this gene in an elegant in vivo shRNA screen for genes that facilitate lung metastasis of 4T1 mouse mammary carcinoma cells: depletion of Akr1B8 was found to reduce lung metastatic burden and similar results are reported for several known metastasis-promoting genes, thereby validating the approach. The investigators report consistent results upon depletion of the human orthologue AKR1B10 in breast cancer lines that express high levels, & reciprocal data upon overexpression of AKR1B10 in human breast cancer cells that express low levels of this gene. BC cells expressing high levels of AKR1B10 exhibit reduced glycolysis and elevated fatty acid oxidation, survive better in low glucose and have reduced evidence of cytotoxic lipid oxidation in low glucose. Etoximir treatment of mice injected with mammary tumour cells eliminates the apparent lung colonization advantage conferred by overexpression of AKR1B10.

While the potential identification of AKR1B10 as a metastasis promoter does appear to be novel, this is not the first study to suggest that oxidative stress limits metastasis (eg. Piskounova et al. Nature 2015) and the mechanism by which AKR1B10 achieves the metabolic switch from glycolysis to FAO is not addressed. Moreover, I have substantial reservations about the quality of analysis and interpretation of the results.

We thank the reviewer for their comments. As detailed below, we have addressed in full all of the reviewer's concerns regarding the analysis and interpretation of the results including conducting additional experiments and further quantification of the metastasis experiments. We have also addressed their comment about novelty and mechanism below (please see response to Major Comment 6). We hope that the reviewer finds this revised submission addresses their concerns.

Major concerns:

1) I am not convinced that measuring total lung weight is an accurate reflection of lung metastatic burden. Besides variation from one mouse to the next, influx of blood during sacrifice can significantly alter tissue mass, especially in the lung. Although supposedly representative images are provided, lung tumour nodule distribution is never even throughout the tissue. A better way to quantify tumour burden is to section through the entire tissue, measuring tumour area every 100 or so microns. This concern is somewhat offset by the luminescence measurements, but the authors need to better calibrate tumour burden with luminescence.

We have addressed this comment in detail as follows:

Fig. 1d (4T1 intravenous inoculation). We have retrieved the lung blocks and cut an additional 2 sections ~150 microns apart. All slides were scanned and quantified for % tumour area in the lung in each section (see new panel in Fig. 1d). These data show a significant decrease in metastatic burden between the shCTRL group and the two shAkr1b8 groups ($p < 0.01$ in both cases), a significant difference between the shRNA non-targeting group (shNTC) and shAkr1b8-4 ($p < 0.05$) and a trend between the shNTC and shAkr1b-7 groups ($p = 0.097$)

Fig. 6e (MDA-MB-231 intravenous inoculation, mice treated with vehicle or etomoxir). As described above, we cut 2 additional sections, ~150 microns apart and quantified the % tumour burden in the lungs (see new panel in Fig. 6b). These data confirm our original finding i.e. that the increased colonisation of AKR1B10^{High} cells is effectively impaired by etomoxir treatment. Of note, due to lack of sufficient material in the tumour blocks we did not perform additional quantification of the experiment shown in Fig. 2d but we state in the text (Page 7) that this experiment is repeated in Fig. 6e (i.e. the comparison mice inoculated with vector-alone and AKR1B10 transfected MDA-MB-231 cells, and treated with vehicle alone (water).

Fig. 1e (4T1 spontaneous metastasis assay). We have repeated the experiment to include an additional shNTC control and an independent Akr1b8 shRNA knockdown cell line (shAkr1b8-7). We quantified the metastatic burden in sections cut at 4 different levels, ~150 microns apart, through the lungs. These new data are provided in a revised Fig. 1e.

2) In Fig. 1d, the effect of Akr1b8 depletion is only significant against the empty vector control but not significant against the more appropriate control, which is non-targeting shRNA. This needs to be addressed

As described in response to Comment 1 above, we have cut additional sections through the lungs blocks to more fully quantify the metastatic burden. As shown in the new Fig. 1d (3rd panel), this quantification shows a significant decrease in metastatic burden between the shCTRL group and the two shAkr1b8 groups ($P < 0.01$ in both cases), a significant difference between the non-targeting shRNA (shNTC) group and shAkr1b8-4 ($P < 0.05$) and a trend between the shNTC and shAkr1b-7 groups ($P = 0.097$).

3) As the authors point out, tail vein-injection is at best an incomplete measure of metastatic potential, which makes the data shown in Figure 1E of paramount importance, given that this is the only measure of spontaneous metastasis in the manuscript. A significant reduction in the number of [lung metastasis] nodules is reported but as discussed in point 1 above, this appears to be an incomplete measurement. Additionally, these data need to be confirmed with a second hairpin and compared with an shNTC control instead of empty vector.

For the spontaneous metastasis experiment shown in Fig. 1e, we repeated the entire experiment with cells transduced with an empty vector control (shCTRL), a non-targeting shRNA (shNTC) and cells transduced with the two independent Akr1b8-targeting shRNA constructs, shAkr1b8-4 and shAkr1b8-7. Cells were inoculated

orthotopically (intramammary fat pad) into BALB/c mice. Primary tumour volume was monitored from day 11 onwards. At the end of the experiment we (a) measured primary tumour weight, (b) collected arterial blood via cardiac puncture and quantified circulating tumour cell colonies after 14 days in culture, (c) assessed the lung metastatic burden in 4 sections cut through each set of lungs and. As metastatic nodules in a spontaneous metastasis assay are smaller than in an experimental intravenous assay, we quantified tumour burden by counting the number of nodules. Representative images of these lungs sections are shown in new Supplementary Figure 2. As shown in new Fig. 1e, this repeated experiment is consistent with our original results and demonstrates:

- (i) no significant difference in primary tumour weight at necropsy between any of the groups
- (ii) a significant reduction in the number of spontaneous metastases in the shAkr1b8-4 and shAkr1b8-7 groups both when compared to the shNTC group (both $p < 0.001$) and compared to the shCTRL group ($p < 0.01$ and $p < 0.05$, respectively). There was no significant difference in number of spontaneous metastases between the shCTRL and shNTC groups or between the shAkr1b8-4 and shAkr1b8-7 groups

In addition, as presented in new Fig. 1f, there was no significant difference between any of the groups in the number of circulating tumour cells, as monitored by tumour cell colonies after culturing arterial blood samples for 14 days.

4) In Figure 2, the analysis of AKR1B10 protein expression in human cell lines is incongruent with the analysis of mRNA expression in breast cancer subtypes. This raises a major concern as to the relevance of human expression data as presented. Does mRNA not reflect protein expression? For publication in a Nature family journal I would really expect TMA analysis of protein expression here.

We believe there is a misunderstanding here and we acknowledge that our original text could have been clearer. As illustrated in Fig. 2a (expression in the TCGA dataset) and Supplementary Fig. 3a (expression in breast cancer cell lines, Neve et al., 2006) expression of *AKR1B10* is, overall, lower in ER+/luminal breast cancers and cell lines than in HER2+/HER2-enriched or ER-/basal-like breast cancers and cell lines. This is fully in keeping with the Western blot data shown in Fig. 2b and the accompanying text where we described low levels of AKR1B10 protein in the ER+ MCF7 and ZR75.1 cell lines and higher levels in the basal-like BT20, MDA-MB-468 and HCC1395 cell lines. However, within these overall expression levels there are examples of ER-/basal-like tumours with low level *AKR1B10* expression (Fig. 2a) and, similarly in the Neve et al. dataset, examples of ER-/basal-like cell lines such as MDA-MB-231 and MDA-MB-453 lines with low level *AKR1B10* expression (Supplementary Fig. 3a).

That said, as the HCC1395 cell line was not present in the Neve et al., profiling set and to confirm these published data we have performed a RTqPCR analysis of all the cell lines shown in Fig. 2b. These new data are presented in Supplementary Fig. 3b so that they can be directly compared to the Neve et al., cell line profiling data. Together these data demonstrate that our western blot data (Fig. 2b) are entirely consistent with the published gene expression profiling data of breast cancer cell lines and that *AKR1B10* mRNA levels do reflect levels of AKR1B10 protein. We have

amended the text on Pages 6 - 7 to clarify this issue and to incorporate the new data presented.

5) The influence of AKR1B10 levels on lung colonization is only shown for 1 cell line (with the same concerns about metastasis burden as raised in [1]) and may only reflect a subclonal difference. Moreover, analysis of spontaneous metastasis needs to be included.

This is not correct. We have shown the influence of AKR1B10 levels on lung colonisation for two cells lines - mouse 4T1 cells where we show reduced lung metastasis when we downregulated *Akr1b8* expression (Fig. 1d, Fig, 1e) and human MDA-MB-231 cells where we show increased lung metastasis following ectopic expression of AKR1B10 (Fig. 2d and Fig. 6b). If the reviewer is referring to the short-term lung colonisation assay, again this has been shown with both 4T1 cells (Fig. 1h) and MDA-MB-231 cells (Fig. 2f).

As described above we have, as requested, repeated the spontaneous metastasis assay to include an additional control and an additional *Akr1b8* knockdown cell line. These new data, which now include an analysis of circulating tumour cells, are presented in Fig. 1e and Fig. 1f.

6) There is no attempt to uncover the mechanism underlying the metabolic switch achieved upon AKR1B10 overexpression

In their introductory comments the reviewer stated "While the potential identification of AKR1B10 as a metastasis promoter does appear to be novel, this is not the first study to suggest that oxidative stress limits metastasis (eg. Piskounova et al. Nature 2015) and the mechanism by which AKR1B10 achieves the metabolic switch from glycolysis to FAO is not addressed."

We agree with the reviewer that this is not the first study to suggest that oxidative stress limits metastasis and in our original manuscript we discussed this and referenced a 2018 review in Cancer Cell. That said, we acknowledge that this part of the manuscript should have been more fully referenced - this has been addressed in the revised manuscript including referencing the Piskounova et al. manuscript (Page 13). However, as the reviewer is aware, how cells combat oxidative stress, particularly in the metastatic microenvironment remains an active research area and to date the role of fatty acid oxidation (FAO) in this process is relatively understudied. The novelty of our study is to identify, via *in vivo* RNAi screening, AKR1B10 as a key player in promoting the survival of metastasising cells within the metastatic microenvironment and to provide mechanistic evidence for a role of AKR1B10 in limiting the toxic side effects of oxidative stress and maintaining increased FAO.

We somewhat disagree with the comment that we have not attempted to uncover the mechanisms underlying the altered metabolism in AKR1B10^{High} cells but we are aware that we should have articulated better the conclusions of our mechanistic data - namely - in the metabolically challenging conditions experienced by metastasising cancer cells, when FAO is required for ATP production, FAO oxidation is limited by the elevated ROS levels associated with oxidative stress/loss of attachment. Our mechanistic data demonstrate that increased expression of AKR1B10 limits the accumulation of lipid peroxides under metabolically challenging conditions

which in turn will limit both oxidative stress-associated cell toxicity and inhibition of FAO. Most importantly, our *in vivo* data support these findings by demonstrating that treatment of mice with the FAO inhibitor etomoxir has no impact on the metastasis of AKR1B10^{Low} cells but inhibits the enhanced metastasis of AKR1B10^{High} cells, directly demonstrating a role for AKR1B10 in maintaining FAO in the metastasising tumour cells and the dependence on FAO for successful metastatic colonisation. In the revised manuscript, we have clarified these issues (Pages 10, 13 - 14).

Minor Concerns:

1) What is the scale on the ordinate axes in Figure 2A?

Apologies - we should have stated on the figure that these are log₂ values. The y axes on the figures and the figure legend has been amended

2) For Figure 4C, False discovery rates should be presented in lieu of p values, as the analysis requires correction for multiple comparisons.

False discover rates are now presented in Fig. 4c. As FDR analysis involves multiple comparison, that data that was originally presented in Fig. 4c and the original Supplementary Fig. 4 is now presented together in the revised Fig. 4c.

3) The scale on the ordinate axes of Figures 1D and 6B should start at 0 rather than 10³

We understand the reviewer's point but as non-tumour bearing lungs have a background IVIS measurement above 0, redrawing the graphs as requested would not be appropriate. However, to clarify the data we have inserted a dotted line into the images representing either the mean IVIS signal or the mean lung weight obtained in age-matched non-tumour bearing mice.

Reviewer #2

The manuscript by Van Weverwijk et al identifies a metabolic enzyme, AKR1B10 as a novel enhancer of breast cancer metastasis from an *in vivo* shRNA screen. The authors demonstrate the following:

1. Depletion of AKR1B10 leads to significant decrease of metastasis to the lung in an intravenous and spontaneous metastasis assay without any effect on the primary tumor growth.
2. Cell lines with high AKR1B10 levels are better at forming tumors in the lungs compared to cells with low AKR1B10 levels in an intravenous metastasis assay, while the *in vitro* colony formation capacity is not affected by AKR1B10 levels.
3. Higher levels of AKR1B10 reduce the glycolytic capacity, glucose uptake and dependency of cells on glucose for survival and proliferation.
4. Increase in AKR1B10 may lead to an increase in fatty acid oxidation, although this point is not made as clearly as the rest
5. High levels of AKR1B10 are correlated with an increase of intracellular fatty acid stores and a decrease in lipid peroxidation
6. High levels of AKR1B10 sensitize metastasizing cancer cells to inhibition of fatty acid oxidation by etomoxir *in vivo*

Specific Comments:

The main finding of the manuscript is very interesting and important to the field of metastasis and metabolic plasticity of metastasizing cancer cells. The fact that cancer cells are able to adapt to their sites of growth during metastasis is an idea that has been gaining traction in the field. Even more importantly, metabolic adaptations during metastasis have been widely discussed in reviews, however not many specific targets that are important for metastatic spread but not primary tumor growth have been identified, which makes this work novel and important. In particular, the ability of etomoxir to block metastasis of AKR1B10-expressing cells is a very impressive and clinically relevant result!

However, there are several concerns that need to be addressed:

1. Throughout the manuscript, authors use lung weight as well as number of nodules that is determined by difference H&E staining to look at the difference in metastasis. However, the cell lines used in the manuscript are labelled with luciferase. It would be more accurate to quantify metastatic burden using bioluminescence, since in several figures including Fig 1e the difference in the number of nodules is very small, and barely statistically significant and seems to be mostly driven by two mice out of 19 that were analyzed. A more accurate quantification of changes in metastatic burden would make the phenotype a lot more convincing. It would be interesting to see if there is metastasis to other organs other than the lung but not necessary for the main point of the manuscript.

As detailed in our response to Reviewer 1 we have repeated the entire experiment shown in Fig. 1e to include an additional control (i.e. non-targeting shRNA) and an additional *Akr1b8* knockdown cell line. In addition, we have performed extensive quantification of metastatic burden by cutting sections through the lungs at 4 different levels. The reason for not performing IVIS imaging in the original or repeated experiment is that the amount of metastatic burden in a spontaneous metastasis assay is low compared to lung tumour burden following intravenous inoculation and, as a consequence, IVIS imaging in spontaneous metastasis assays does not provide an accurate measure of tumour burden in the metastatic sites. We hope that this repeated experiment with additional control and a more thorough quantification of the metastatic nodules in the lungs at 4 different levels addresses the Reviewer's concern.

With regards to metastasis to other organs, we did examine the brains and livers of mice in the repeated spontaneous metastasis assay. As expected (and seen by other groups), there is very limited colonisation of other metastatic sites by 4T1 cells following orthotopic inoculation. We agree with the Reviewer that in future experiments it would be of interest to examine the role of increased AKR1B10 levels at other metastatic sites, but this is beyond the scope of the current study.

2. The authors want to address the effect of *Akr1b8*/AKR1B10 knockdown on survival of cancer cells in circulation (for examples in Fig 1f), however the assay used in the manuscript is not convincing and not correct. Mixing the cells with the dye and injecting them in the tail vein to see how many cells end up lodged in the lung and how many remain after 8 or 16 hours is not modeling survival in circulation, but rather most likely

looking at the clearing of the cells from the lung as well as their survival in the lung, since cells go straight from the injection to the lung within seconds with minimal time in circulation. The authors have a spontaneous metastasis model that they use in the manuscript already, it would be much more convincing and accurate to look for the presence of cancer cells in the blood of those mice by flow cytometry.

The reviewer is correct and we apologise that we were not clearer in our writing. The "lung retention" assay that we used monitors survival of tumour cells that are still within the circulatory system (i.e. have not yet extravasated into the lung parenchyma) but these cells will have rapidly lodged within the lung capillaries, hence they are not freely distributed within the circulation. We have amended the text of the manuscript to state this more accurately (Pages 6 and 7). More importantly we have performed two additional experiments

(i) As suggested by the reviewer, performed a more direct assay to monitor circulating tumour cells when we repeated the spontaneous metastasis assay from Fig. 1e. As shown in the revised figure, we collected arterial blood at the end of the experiment and after 14 days in culture counted the number of tumour cell colonies. As shown in new Fig. 1f, there was no significant difference in number between or within the control groups and the two *Akr1b8* knockdown group.

(ii) As an indirect assay, we have performed *in vitro* anoikis assays assessing the apoptosis of tumour cells cultured in non-adherent conditions. As shown in new Fig. 1g and new Fig. 2e, we found no significant difference in apoptosis between either shNTC and sh*Akr1b8* 4T1 cells or between MDA-MB-231 AKRB10^{High} and AKR1B10^{Low} cells

3. Authors use confluency fold change in Figure 3 as a way to monitor effects of glucose depletion on cell growth. This is an unusual way to represent the results. It would be better if the data was confirmed with an alternative approach such as changes in cell number or cell survival (e.g. with Cell Titer Glo), which the authors use in other figures in the manuscript already.

The data presented in Fig 3c comes from monitoring live cells continuously over 4.5 days using an IncuCyte. Using the IncuCyte avoids having to plate out multiple replicates for measuring cell viability/number using techniques such as CellTiter-Glo at fixed time points. As a consequence, the IncuCyte is now commonly used by researchers in many different fields, as evidenced by the many publications in different journals including Nature Communications (for two examples see <https://www.nature.com/articles/ncomms15522#methods> <https://www.pnas.org/content/114/15/3933#sec-16>). The output from the IncuCyte continuous phase contrast imaging is % confluency in the well and we have expanded on this in the Methods section (Page 19 - 20). In summary, this is not an unusual technique, rather it is an increasingly routine method for monitoring cell growth.

4. Authors clearly show that they have both overexpression and knockdown cell lines for AKR1B10. The manuscript would much stronger and more convincing if the authors used both sets of cell lines for each of the assays shown, rather than pick and choose which one to show.

The main part of the manuscript where we did not analyse all three human cell lines was in the growth assays (Fig. 3c - e, Fig. 5a) and this is because HCC1395 cells have a very slow proliferation rate. In addition, HCC1395 cells do not form tumours when inoculated into mice and therefore cannot be used for *in vivo* experiments. However, to provide additional validation to our studies, we have repeated the experiments in Fig. 5c and 5e (i.e. quantification of lipid storage and lipid peroxidation) with the HCC1395 (which have high endogenous AKR1B10 levels) transfected with siNTC and siAKR1B10 oligonucleotides and obtained equivalent results for AKR1B10^{High} and AKR1B10^{Low} cells as for the MDA-MB-231 AKR1B10^{Low} cells transfected with empty vector (Vec) or an AKR1B10 construct. These new data are shown in Fig. 5c and Fig. 5e.

Minor comments:

1. Why are there are no error bars on the control bars in Figure 2e?

We think the reviewer is referring to Figure 3e. In Fig. 3e we originally expressed the data relative to the 4.5 g/L samples, hence the 4.5 g/L samples are set at 1.0 and do not have error bars. In the revised Fig. 3e the data is presented as absolute values (mm²). This has no effect on the findings i.e. there is no significant difference between AKR1B10^{High} and AKR1B10^{Low} samples cultured in high glucose (4.5 g/L) but a significant reduction in AKR1B10^{Low} colony growth in low glucose conditions.

2. Figure 4 is probably the main point of the paper since it shows effects of AKR1B10 on Fatty Acid Oxidation but it is very difficult to follow even for someone familiar with the assays. Better labeling and schematics would really help.

3. Generally better/more precise labelling of the axes and figures (e.g. Figure 4 and 5) would make the manuscript a lot easier to read.

Apologies, on review we appreciate that we could have been much clearer in our labellings and descriptions. To address this we have, in Figures 4 and 5, (a) changed the y axis labellings, (b) provided headings for the different panels, (c) revised the relevant Methods section, making sure that the calculations are clearly stated. We have also revised the figure legends and main text. We also appreciate that our presentation of the glutamine and fatty acid dependency data was particularly confusing. In the revised manuscript these data are now presented together in Fig. 4a. We hope these changes provide greater clarity to the assays undertaken.

Overall, this is a very interesting paper with a lot of convincing and impressive results. It could be a great contribution to the field once the major concerns are addressed.

We thank the reviewer for their comments and we hope that the additional experiments, further quantification of the metastasis data and improvements to the text/figures/figure legends have addressed their concerns.

Reviewers' Comments:

Reviewer #1:

Remarks to the Author:

Having read the revised manuscript, I agree that the authors have addressed many of my concerns and that the amendments to the text have indeed improved the clarity of the message. I do however continue to have some specific reservations that I imagine can be easily addressed:

1) Figure 1D. I acknowledge the inclusion of a second hairpin, however the lung weight and bioluminescence data remain not significantly different from the most appropriate control (shNTC) and should therefore be omitted. As regards to the tumour area data, does a difference in the efficiency of Akrb8 protein depletion account for the failure of the second hairpin to significantly impact lung colonisation?

2) Figure 1E. The disparity in the number of data points comparing the primary tumour weight panel (center) with the tumour nodule enumeration panel (right) suggests that the right panel shows nodules per tissue section rather than per mouse. This artificially inflates the statistical power and is poor practice. The panel should be redrawn as nodules per mouse, averaged across 4 sections, and the statistical significance determined on a per mouse basis not a per section basis.

3) Figure 2D. The authors persist with using lung weight here as a measure of tumour burden. As indicated in the original review, this is an unreliable measure and should be replaced with tumour area, presented as discussed for Figure 1E above. This point is underlined by the lack of significance of AKR1B10^{high} untreated versus etoxomir-treated lung weights presented in Figure 6B, despite a significant difference in the tumour area from the same samples.

4) Figure 4a. The cellular requirement for pyruvate is mentioned in the text but the data are not shown.

As regards the overall significance of the findings, in my opinion the data presented in Figure 1E are pivotal, as this is the only experiment that addresses spontaneous metastasis formation (and this is only performed with the murine cell line). If, upon correction of the nodule enumeration data as per point 2, the depletion of Akr1B8 fails to significantly impact nodule number, then all of meaningful *in vivo* data will be derived from tail vein injection of 1 murine and 1 human cell line, which is a rather artificial surrogate for metastasis. Under such circumstances, I would be reluctant to recommend publication in a Nature family journal without seeing supporting *in vivo* data using additional human cell lines.

Reviewer #2:

Remarks to the Author:

The authors have addressed all of my concerns and I recommend the manuscript for publication.

One minor suggestion:

1) It would make the manuscript more thorough and complete to show both shRNAs against Akr1b8 in Fig 1 g and h, like in the rest of the panels in Fig 1.

However, this is unlikely to affect the overall conclusion of the paper and therefore should not delay publication.

Metabolic adaptability in metastatic breast cancer by AKR1B10-dependent balancing of glycolysis and fatty acid oxidation

Antoinette van Weverwijk, Nikolaos Koundouros, Marjan Irvani, Matthew Ashenden,
Qiong Gao, George Poulogiannis, Ute Jungwirth and Clare M. Isacke

Response to reviewers' comments:

Reviewer #1 (Remarks to the Author):

Having read the revised manuscript, I agree that the authors have addressed many of my concerns and that the amendments to the text have indeed improved the clarity of the message.

We thank the reviewer for these comments.

I do however continue to have some specific reservations that I imagine can be easily addressed:

In light of the comments made by Reviewer 1 below, we have followed the advice of two independent statisticians as to the most appropriate statistics to use when an experiment contains two control groups (in this case the shNTC and shCTRL groups) and two experimental groups (shAkr1b8-4 and shAkr1b8-7 groups). Following their advice we have used one-way ANOVA followed by the two-stage step-up method of Benjamini, Krieger and Yekutieli for comparisons between an individual control group and an individual experimental group (i.e. in Figure 1d, Figure 1e and Figure 1f). This is now stated in the Methods section.

1) Figure 1D. I acknowledge the inclusion of a second hairpin, however the lung weight and bioluminescence data remain not significantly different from the most appropriate control (shNTC) and should therefore be omitted. As regards to the tumour area data, does a difference in the efficiency of Akrb8 protein depletion account for the failure of the second hairpin to significantly impact lung colonisation?

There is a minor confusion here. In the first revision of our manuscript we did not include a second hairpin in Figure 1d - data using two independent hairpins had always been shown (i.e. they were present in our original submission). The only change to Figure 1d made in the first revision was, as requested by the reviewer, the inclusion of the third panel i.e. the quantification of lung tumour burden where we cut additional sections through the lungs and quantified the metastases by measuring the mean % tumour area in the lungs (average of three separate sections).

That said, using the recommended statistics as described above, individual comparisons of metastatic burden (as monitored by IVIS signal, lung weight and immunohistochemical quantification of metastasis) between a control and an experimental group are all significantly significant except for Figure 1e where the comparison between the shCTRL and shAkr1b8-7 groups has a \$P\$ value of 0.058.

2) Figure 1E. The disparity in the number of data points comparing the primary tumour weight panel (center) with the tumour nodule enumeration panel (right) suggests that the right panel shows nodules per tissue section rather than per mouse. This artificially inflates the statistical power and is poor practice. The panel should be redrawn as nodules per mouse, averaged across 4 sections, and the statistical significance determined on a per mouse basis not a per section basis.

The quantification of metastatic burden in the spontaneous metastasis assay (Figure 1e) is, as requested, now presented as mean number of lung nodules per mouse (averaged over 4 sections). Details of the statistics are described above.

3) Figure 2D. The authors persist with using lung weight here as a measure of tumour burden. As indicated in the original review, this is an unreliable measure and should be replaced with tumour area, presented as discussed for Figure 1E above. This point is underlined by the lack of significance of AKR1B10high untreated versus etoxomir-treated lung weights presented in Figure 6B, despite a significant difference in the tumour area from the same samples.

We agree with the reviewer that it is not ideal to only show lung weight as a monitor of tumour burden but as we explained in our original rebuttal level (a) we were unable to provide histological quantification of this experiment due to lack of lung tissue remaining in the block, and (b) this experiment was repeated in Figure 6b where we compared vehicle-alone treated mice. However, as this remains a sticking point with the referee, we have removed this panel and simply, in the main text, refer the reader to the vehicle-treated samples presented in Figure 6b.

4) Figure 4a. The cellular requirement for pyruvate is mentioned in the text but the data are not shown.

We apologise, we should have been clearer in our writing. We make no claims about a cellular requirement for pyruvate. The confusion arises in that to monitor glutamine dependency using the Mito Fuel Flex Test, you first inhibit glutamine import and then inhibit import of both fatty acids and pyruvate. Similarly to monitor fatty acid dependency, you first inhibit fatty acid import and then inhibit glutamine and pyruvate import. This was clearly stated in the Methods section but was not so clear in the text or figure legends. We have now clarified these sections.

As regards the overall significance of the findings, in my opinion the data presented in Figure 1E are pivotal, as this is the only experiment that addresses spontaneous metastasis formation (and this is only performed with the murine cell line). If, upon correction of the nodule enumeration data as per point 2, the depletion of Akr1B8 fails to significantly impact nodule number, then all of meaningful in vivo data will be derived from tail vein injection of 1 murine and 1 human cell line, which is a rather artificial surrogate for metastasis. Under such circumstances, I would be reluctant to recommend publication in a Nature family journal without seeing supporting in vivo data using additional human cell lines. We agree that the data shown in Figure 1e is important and we trust the reviewer is now satisfied the data we present, particularly the measure of mean tumour nodules per mouse averaged over 4 sections.

Reviewer #2 (Remarks to the Author):

The authors have addressed all of my concerns and I recommend the manuscript for publication.

One minor suggestion:

1) It would make the manuscript more thorough and complete to show both shRNAs against Akr1b8 in Fig 1 g and h, like in the rest of the panels in Fig 1.

However, this is unlikely to affect the overall conclusion of the paper and therefore should not delay publication.

We thank the reviewer for recommending the manuscript for publication. We agree that for completeness it would be ideal to show all of the cell lines in Figure 1g and 1h but we believe that this will not add significantly to the manuscript and we are reluctant to perform additional animal experiments without a strong justification. We appreciate the reviewer stating that this is not essential for publication.

Reviewers' Comments:

Reviewer #1:

Remarks to the Author:

The authors have addressed my outstanding concerns and I am now satisfied that the manuscript is suitable for publication and congratulate the authors on a nice study.